# Genomewide phenotypic analysis of growth, cell morphogenesis, and cell cycle events in *Escherichia coli*

Manuel Campos[1,2,3,4], Sander K Govers[1,2], Irnov Irnov[1,2,†], Genevieve S Dobihal[1,3,‡], François Cornet[4] & Christine Jacobs-Wagner[1,2,3,5,*] (iD)

## Abstract

Cell size, cell growth, and cell cycle events are necessarily intertwined to achieve robust bacterial replication. Yet, a comprehensive and integrated view of these fundamental processes is lacking. Here, we describe an image-based quantitative screen of the single-gene knockout collection of *Escherichia coli* and identify many new genes involved in cell morphogenesis, population growth, nucleoid (bulk chromosome) dynamics, and cell division. Functional analyses, together with high-dimensional classification, unveil new associations of morphological and cell cycle phenotypes with specific functions and pathways. Additionally, correlation analysis across ~4,000 genetic perturbations shows that growth rate is surprisingly not predictive of cell size. Growth rate was also uncorrelated with the relative timings of nucleoid separation and cell constriction. Rather, our analysis identifies scaling relationships between cell size and nucleoid size and between nucleoid size and the relative timings of nucleoid separation and cell division. These connections suggest that the nucleoid links cell morphogenesis to the cell cycle.

**Keywords** cell shape; cell size; Keio library; nucleoid separation; nucleoid size

**Subject Categories** Genome-Scale & Integrative Biology; Methods & Resources; Microbiology, Virology & Host Pathogen Interaction

**Mol Syst Biol. (2018) 14: e7573**

## Introduction

Cells must integrate a large variety of processes to achieve robust multiplication. Bacteria, in particular, are remarkable at proliferating, which has been key to their colonization success. During their fast-paced replication, bacterial cells must uptake and process nutrients, generate energy, build cellular components, duplicate and segregate their genetic material, couple growth and division, and maintain their shape and size, all while sensing their environment and repairing cellular damages, just to name a few important tasks. These tasks must be integrated to ensure successful cellular replication. Decades of work have garnered extensive knowledge on specific processes, genes, and pathways, but we still lack a comprehensive view of the genetic determinants affecting cell morphogenesis and the cell cycle. It is also unclear how cellular activities are integrated to ensure that each division produces two viable daughter cells.

Systematic genomewide screens, rendered possible by the creation of arrayed single-gene knockout collections, have been successfully used to gain a more comprehensive perspective on cell morphogenesis and the cell cycle in yeast (Jorgensen *et al*, 2002; Ohya *et al*, 2005; Graml *et al*, 2014). Beyond the functional information gained through the mapping of phenotypes associated with the deletion of genes, genomewide screens also provide a unique opportunity to interrogate the relationship between phenotypic features with thousands of independent genetic perturbations (Liberali *et al*, 2015). Here, we present a high-content, quantitative study that uses the Keio collection of *Escherichia coli* gene deletion strains (Baba *et al*, 2006) and combines microscopy with advanced statistical and image analysis procedures to examine the impact of each non-essential *E. coli* gene on cell morphology, cell size, growth, nucleoid (bulk chromosome) dynamics, and cell constriction. In addition, we provide insight into the connectivity and empirical relationships between cell morphogenesis, growth, and late cell cycle events.

1  Microbial Sciences Institute, Yale University, West Haven, CT, USA
2  Department of Molecular, Cellular and Developmental Biology, Yale University, New Haven, CT, USA
3  Howard Hughes Medical Institute, Yale University, New Haven, CT, USA
4  Laboratoire de Microbiologie et Génétique Moléculaires (LMGM; UMR5100), Centre de Biologie Intégrative (CBI), Centre National de la Recherche Scientifique (CNRS), Université de Toulouse, UPS, Toulouse, France
5  Department of Microbial Pathogenesis, Yale School of Medicine, New Haven, CT, USA
   *Corresponding author. Tel: +1 203 737 6778; Fax: +1 203 737 6715; E-mail: christine.jacobs-wagner@yale.edu
   †Present address: Department of Microbiology, New York University School of Medicine, New York, NY, USA
   ‡Present address: Department of Microbiology and Immunology, Harvard Medical School, Boston, MA, USA

# Results

### High-throughput imaging and growth measurements of the *E. coli* Keio collection

To gain an understanding of the molecular relationship between growth, cell size, cell shape, and specific cell cycle events, we imaged 4,227 strains of the Keio collection. This set of single-gene deletion strains represents 98% of the non-essential genome (87% of the complete genome) of *E. coli* K12. The strains were grown in 96-well plates in M9 medium supplemented with 0.1% casamino acids and 0.2% glucose at 30°C. The preferred carbon source

(glucose) and the casamino acids provide growth conditions that give rise to overlapping DNA replication cycles (Appendix Fig S1A). Live cells were stained with the DNA dye DAPI and spotted on large custom-made agarose pads (48 strains per pad) prior to imaging by phase-contrast and epifluorescence microscopy (Fig 1A). On average, about 360 (±165) cells were imaged for each strain. To provide a reference, 240 replicates of the parental strain (BW25113, here referred to as WT) were also grown and imaged under the same conditions as the mutants. In parallel, using a microplate reader, we recorded the growth curves of all the strains (Fig 1A) and estimated two population-growth features. We fitted the Gompertz function to estimate the maximal growth rate ($\alpha_{max}$) and used the last hour of

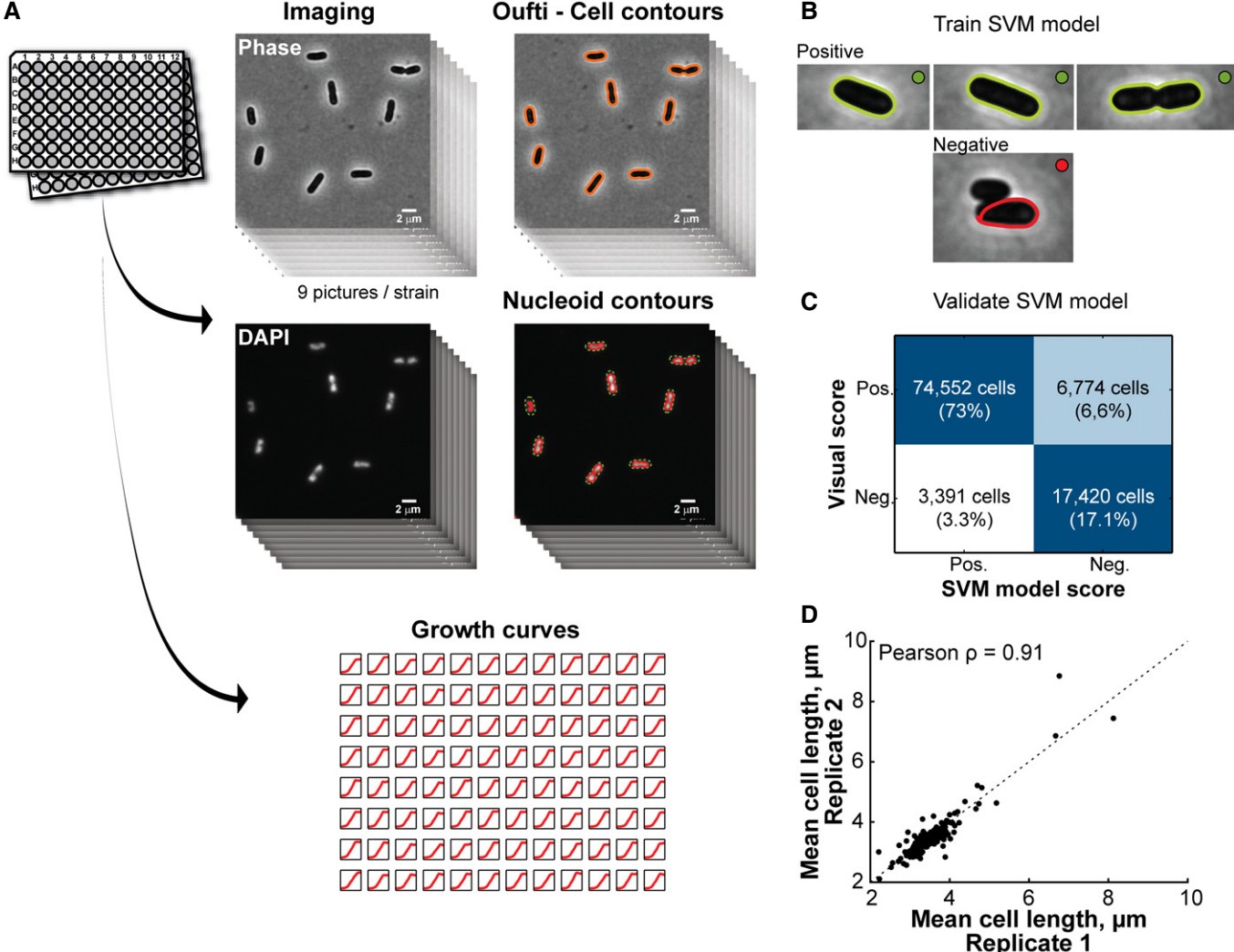

**Figure 1. Experimental approach and reproducibility.**

A  Experimental workflow. Single-gene knockout strains from the Keio collection were grown in M9-supplemented medium at 30°C in 96-well plates. DNA was stained with DAPI prior to imaging, and nine images were taken in both phase-contrast and DAPI channels. The images were then processed with MicrobeTracker and Oufti to identify the cell and nucleoid contours. In parallel, we recorded the growth curve of each imaged strain in order to extract growth parameters.

B  A SVM model was trained via visual scoring of 43,774 cells.

C  Confusion matrix of the SVM model based on a large validation dataset (102,137 cells), illustrating the distribution of the SVM classifier output in comparison with the visual classification.

D  Comparison of the average cell length of 178 strains obtained from two independent 96-well cultures of the 176 most phenotypically remarkable Keio strains and two WT replicates.

growth to calculate the saturating density ($OD_{max}$) of each culture (Appendix Fig S1B). The goodness of fit is illustrated at the time of maximal growth where the $OD_{600\ nm}$ from the growth curve is highly correlated with the $OD_{600\ nm}$ predicted by the fit (Appendix Fig S1C). The vast majority of strains were imaged in exponential phase at an $OD_{600\ nm}$ ($OD_{imaging}$) 4–5 times smaller than their $OD_{max}$ (Appendix Fig S1D).

### High-throughput dataset curation using a support vector machine

Cells and their contours were detected in an automated fashion (Sliusarenko *et al*, 2011). The large size (>1,500,000 detected cells) of the dataset precluded the validation of each cell contour by visual inspection. Therefore, we implemented an automated classification method based on a support vector machine (SVM; Fan *et al*, 2005) to identify and discard incorrectly detected cells (Fig 1B). To generate a training dataset for the SVM model, we visually scored (positive or negative) 43,774 cell contours from the parental strain and the 419 mutants displaying the greatest deviations in cellular dimensions before data curation. The inclusion of the most aberrant mutants in the training dataset allowed us to build a versatile model that performed well on the wide range of cell sizes and shapes present in our dataset. The quality of the fit of the SVM model to the training dataset was evaluated by a 10-fold cross-validation (Hastie *et al*, 2009), which gave a misclassification error rate of 10%. The model was further validated on an independent dataset of 102,137 visually scored cell contours taken from the same group of WT and mutant strains. We found that our SVM model performed very well on this validation set, as shown by the high AUROC (area under the "receiver operating characteristic" curve) value of 0.94 (Appendix Fig S1E). By comparing the model classification with visual scoring (Fig 1C), we found that only about 3% of cell contours in the validation set were incorrectly identified as positive by the SVM model. Importantly, these false-positive cells introduced no biases in the measurement of the SVM predictor values (Appendix Fig S1F), even when considering the 419 most aberrant strains (Appendix Fig S1G). This validated SVM model was used to curate the entire dataset, retaining about 1,300,000 identified cells (291 ± 116 cells/strain). In addition, we verified the reproducibility of our experimental approach by separately imaging two independent replicates of 178 strains that included two copies of the parental (WT) strain and 176 mutants with severe morphological defects. We observed a Pearson correlation (ρ) of 0.91 for cell length (Fig 1D), indicating high reproducibility.

### Quantification of cell morphological features across the genome

With this high-quality dataset, we were able to obtain a wealth of quantitative information using the software packages Microbe-Tracker and Oufti (Sliusarenko *et al*, 2011; Paintdakhi *et al*, 2016). From phase-contrast images, we measured cellular dimensions, such as length, width, perimeter, cross-sectional area, aspect ratio (width/length), and circularity ($4\pi$ area/(perimeter)$^2$). We also measured the variability of these features by calculating their coefficient of variation (CV, the standard deviation divided by the mean). From both series of measurements, we extracted the mean and CV of additional morphological parameters, such as surface area, volume, and surface-to-volume ratio. For constricted cells, we

determined the relative position of division along the cell length (division ratio). Note that since the identity of the cell poles (old versus new) was unknown, randomization of cell pole identity would automatically produce a *mean* division ratio of 0.5, even for an off-center division. Therefore, measurements of mean division ratio were meaningless and not included in our analysis. However, the CV of the division ratio was included since a high CV indicated either an asymmetric division or an imprecise division site selection. In total, each strain was characterized by 19 morphological features (see Dataset EV1 for raw data). The name and abbreviation for all the features can be found in Appendix Table S1.

After taking into consideration experimental variability (see Materials and Methods, Appendix Figs S2–S4), we calculated a normalized score (*s*) for each feature and each strain (see Materials and Methods). The corrected and normalized data (scores) can be found in Dataset EV2. Even with a conservative threshold of 3 standard deviations ($s \leq -3$ or $\geq 3$, or absolute score $|s| \geq 3$) away from the WT, a large number (874) of single-gene deletion strains were associated with one or more morphological defects (Fig 2 and Dataset EV2). This result indicates that a large fraction of the non-essential genome (i.e., ~20% of the unique deletion strains present in the Keio collection) directly or indirectly affect cell size and shape. Similar genomic commitment to cell size and shape was observed in budding yeast (Jorgensen *et al*, 2002).

### Quantification of growth and cell cycle features across the genome

From the images, we also calculated the degree of constriction for each cell and determined the fraction of constricting cells in the population for each strain (see Materials and Methods). From the latter, we inferred the timing of initiation of cell constriction relative to the cell cycle (Powell, 1956; Collins & Richmond, 1962; Wold *et al*, 1994). In addition, the analysis of DAPI-stained nucleoids with the objectDetection module of Oufti (Paintdakhi *et al*, 2016) provided additional parameters, such as the number of nucleoids per cell and the fraction of cells with one versus two nucleoids. From the fraction of cells with two nucleoids, we estimated the relative timing of nucleoid separation (Powell, 1956; Collins & Richmond, 1962; Wold *et al*, 1994). We also measured the degree of nucleoid constriction in each cell for each strain and compared it to the degree of cell constriction to obtain the Pearson correlation between these two parameters, as well as the average degree of nucleoid separation at the onset of cell constriction (Appendix Fig S1H). As a result, each strain was associated with five cell cycle features (Dataset EV2), in addition to the 19 morphological features and two growth features mentioned above (see Appendix Table S1).

We found that 231 gene deletions were associated with at least one dramatically altered ($|s| \geq 3$) cell cycle feature (Fig 2, Dataset EV2). From the growth curves, we identified over 263 mutants with severe ($|s| \geq 3$) growth phenotypes (Fig 2, Dataset EV2) despite the growth medium being supplemented with amino acids.

### Severe defects in growth, cell morphology, or the cell cycle are associated with a wide variety of cellular functions

For each feature, the genes deleted in mutant strains with a $|s| \geq 3$ encompassed a wide range of cellular functions based on a

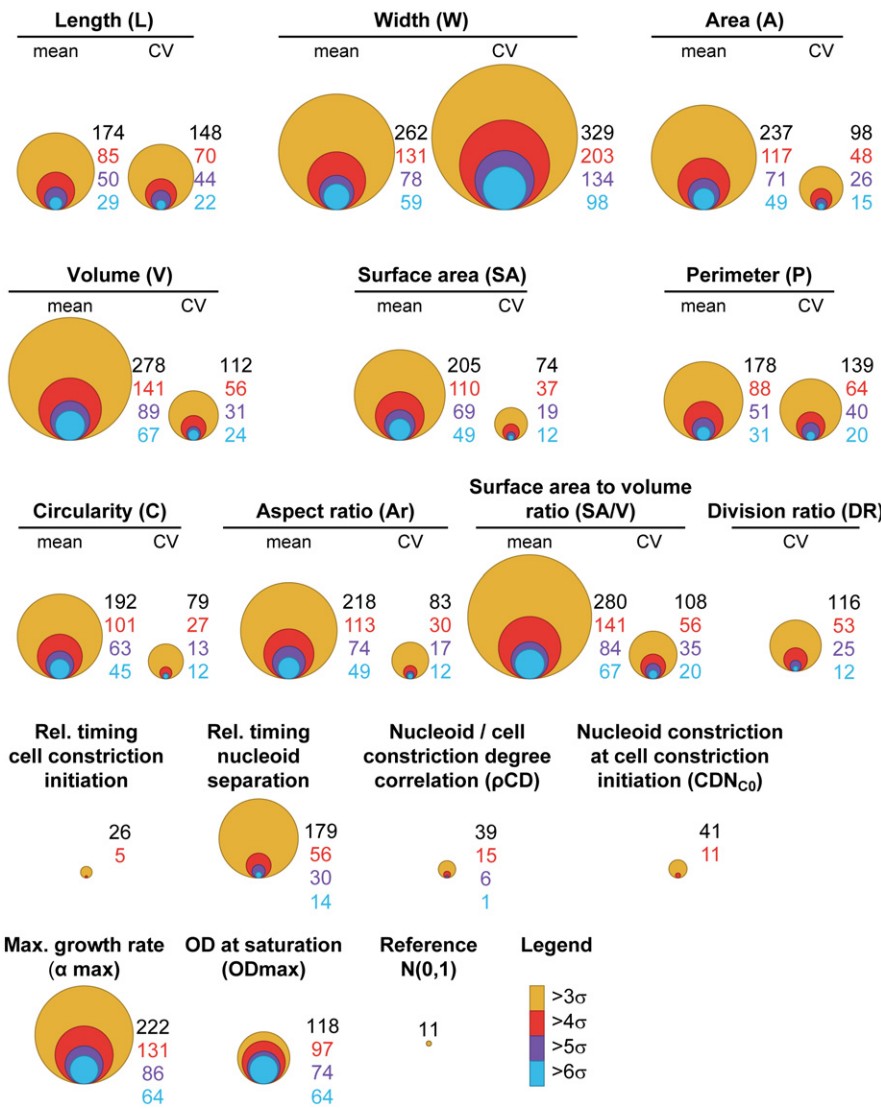

**Figure 2.  Distribution of morphological, cell cycle, and growth phenotypes in the *Escherichia coli* Keio strain collection.**

Bubble graphs representing, for each feature, the number of strains with a score value, *s*, beyond 3, 4, 5, or 6 times the interquartile range away from the median of the WT distribution (240 replicates), corrected by a factor 1.35 to express the deviation in terms of standard deviations (iqr $\approx$ 1.35 × σ for a normal distribution). The size of the circles (bubbles) reflects the number of strains with a score beyond a specific, color-coded threshold for *s*, as indicated. The "reference" bubble graph illustrates the expectations from a dataset of the same size (4,227 strains), assuming a standardized normal distribution of scores (with a mean of 0 and a standard deviation of 1).

distribution analysis of Clusters of Orthologous Groups (COGS) of proteins (Fig 3 and Appendix Fig S5). This diversity highlights the high degree of integration of the cell cycle and cell morphology in overall cellular physiology.

Certain COGs were statistically enriched for some phenotypes (Fig 3). We recovered expected associations, such as category D (cell cycle control, cell division, and chromosome partitioning) with high mean length (<L>) and high length variability (CV$_L$) and category M (cell wall/membrane/cell wall biogenesis) with high mean width (<W>; Fig 3A). Indeed, defects in DNA partitioning and repair can lead to a cell division block (Mulder & Woldringh, 1989), and impairment in cell envelope biogenesis has been reported to cause cell widening (Bean *et al*, 2009; Lee *et al*,

2014). COG categories associated with translation or some aspect of metabolism were, not surprisingly, enriched in mutants with growth defects (Fig 3B). Category H was enriched among small ($s < -3$) mutants. This category encompasses a number of pathways important for general aspects of metabolism (e.g., biosynthesis of pantothenate, electron carriers, biotin, and chorismate), suggesting that their impairment affects cell size in a manner similar to nutritional restriction.

Often, these COG enrichments were carried over to features (area, volume, perimeter, circularity, etc.) that directly relate to width and length (Fig 3). However, we also observed differential COG enrichments even for highly related features, highlighting the importance of considering features beyond mean and CV of length

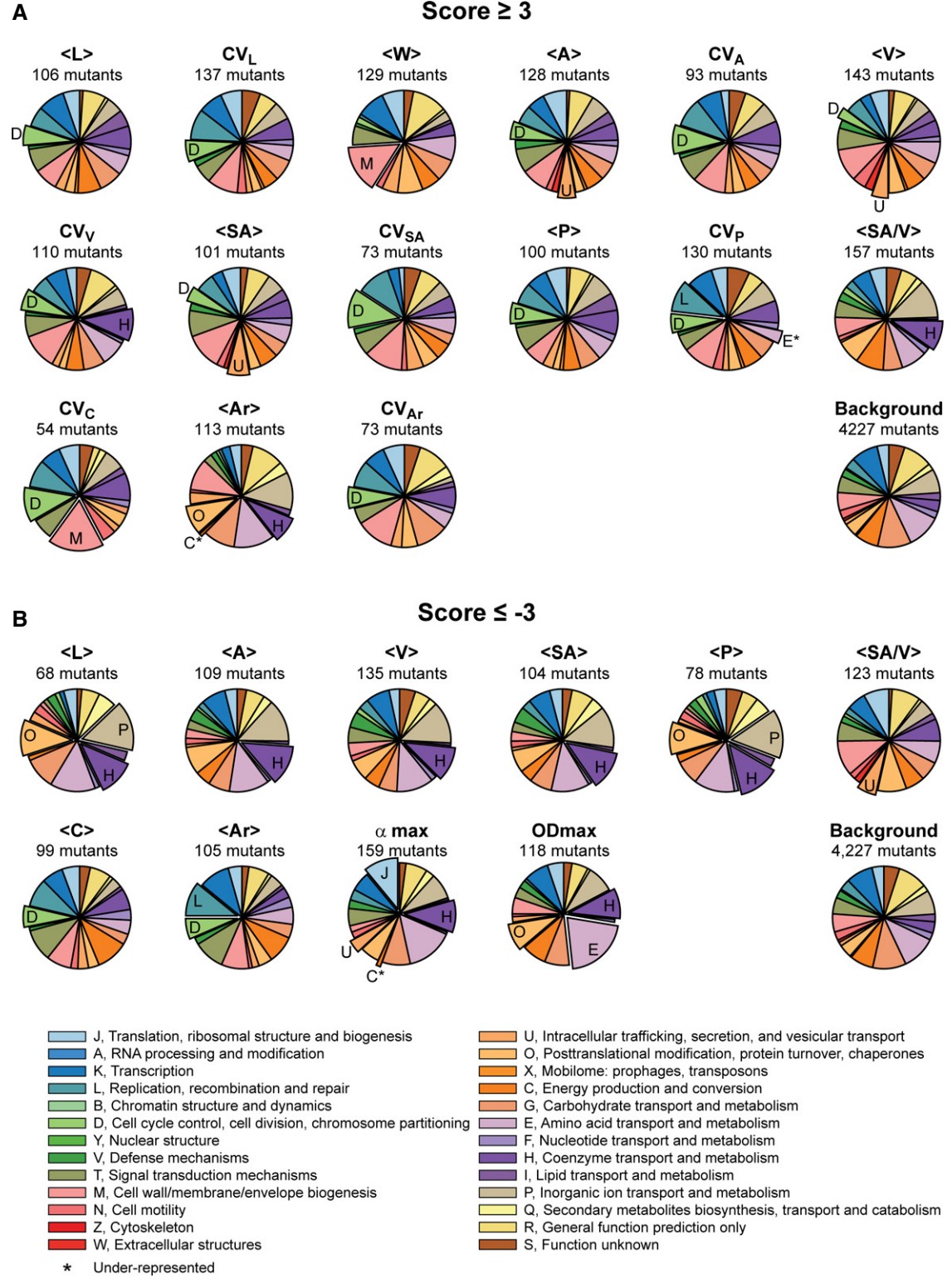

**Figure 3.  Feature-based COG enrichment analysis.**

A   Pie charts representing, on a feature-by-feature basis, the relative distribution of COG categories among the gene deletion strains associated with $s \geq 3$. The enriched COG categories are labeled and highlighted with an exploded pie sector. The under-represented COG categories are further highlighted by an asterisk. Enrichments and under-representations with an associated (FDR-corrected) $q$-value < 0.05 were considered significant. Only morphological and growth features with at least one enriched or under-represented COG category are represented.

B   Same as in (A) but for $s \leq -3$.

and width. For example, category U (intracellular trafficking, secretion, and vesicular transport) was enriched among mutant strains with high mean area (<A>) and volume (<V>) but normal <L> or <W> (Fig 3A), suggesting that small deviations in length and width can combine to produce significant differences in area and volume. On the other hand, deletions in category C genes (energy production and conversion) were normally represented for most phenotypes, but were conspicuously under-represented among mutants with high mean shape factors, suggesting that deletion of these genes was barely associated with a high aspect ratio (<Ar>; Fig 3). Thus, deletion of genes involved in energy and conversion can influence the size of the cell without affecting its shape (aspect ratio and circularity), implying that a defect in length is often accompanied by a defect in width in this category of mutants.

## High-dimensional classification of the morphological mutants

While the gene deletion annotation of the Keio library is not perfect, our large dataset provided a powerful platform to examine global trends and to identify gene function enrichments in phenotypic classes of mutants with $|s| \geq 3$. First, we considered morphological phenotypes. Instead of ranking strains on a feature-by-feature basis, we sought to classify strains based on their combination of features, or "phenoprints", to better capture the phenotypic complexity of morphology. In addition to the 19 morphological features, we included the two growth-related features ($OD_{max}$ and $\alpha_{max}$) in the phenoprint because growth rate is often assumed to affect cell size. This assumption stems from the early observation that bacterial cell size (mean cell mass) scales with growth rate when the latter is modulated by varying the composition of the culture medium (Schaechter *et al*, 1958). This scaling relationship has historically been referred to as the "growth law".

The combination of the 21 scores was used to classify a dataset composed of 240 wild-type replicates (controls) and 985 mutant strains with a $|s| \geq 3$ for at least one morphological or growth feature. A principal component analysis (PCA) of this dataset was not useful, as it identifies strong outliers with severe phenotypes but failed to separate most strains (Appendix Fig S6). Therefore, to identify strains with similar phenoprints, we turned to the machine learning "t-distributed stochastic neighbor embedding" (t-SNE) algorithm (van der Maaten & Hinton, 2008). Unlike PCA, which identifies the principal components that most explain the variance in a dataset, the principle of t-SNE is to minimize distances between datapoints with high mutual information. Thus, t-SNE can be used to emphasize similarities, rather than dissimilarities. Taking advantage of the stochastic nature of t-SNE, we generated 100 independent maps and used the density-based clustering algorithm dbscan (Ester *et al*, 1996) to identify strains that reproducibly (> 90% of the time) clustered together in the t-SNE maps. Using this combined t-SNE-dbscan approach, we found that the wild-type replicates clustered together to form the "WT" island while the mutant strains consistently separated in 22 islands (Fig 4A).

Each island of the "morpho-archipelago" was characterized by an average phenoprint (Fig 4B), with a given feature often segregating in different islands. For example, slowly growing (low $\alpha_{max}$) mutants were found in both islands 1 and 5, but mutants in island 1 were, on average, small whereas mutants in island 5 were morphologically like WT (Fig 4B). Thus, island 5 illustrates a group of

strains that departs from the growth law, as they produce cells that are as large as WT despite growing slower.

## Genes, functions, and pathways associated with cell size and shape

Our t-SNE classification identified many new genes associated with specific phenotypes, even extreme ones. For example, island 22 grouped strains characterized by cells that were very long and highly variable in length (and therefore area, volume, surface area, perimeter, and aspect ratio), but had a comparatively normal width (Fig 4B). Such a cell filamentation phenotype has been well studied, and our classification recovers expected gene deletions, such as Δ*minC*, Δ*envC*, Δ*tatC*, Δ*hfq*, and Δ*dedD* (Figs 4C and EV1A; Adler *et al*, 1967; Rodolakis *et al*, 1973; Tsui *et al*, 1994; Stanley *et al*, 2001; Gerding *et al*, 2009). Island 22 also includes four gene deletions (Δ*uup*, Δ*rdgB*, Δ*croE*, and Δ*ydaS*) that were unknown for their cell filamentation phenotype (Figs 4C and EV1A), suggesting new or unappreciated functions connected to cell division. For example, Uup is a DNA-related protein known to prevent the precise excision of transposons (Hopkins *et al*, 1983). The working model postulates that Uup interacts with the replisome to prevent replication forks stalling at the repeated sequences flanking transposons, a step required for the formation of a Holliday junction and excision (Murat *et al*, 2006). Replisomes also frequently stop at other chromosomal regions during replication, which can cause DNA lesions (Cox *et al*, 2000). If DNA damages are left uncorrected, they lead to inhibition of cell division. The cell filamentation phenotype associated with the deletion of *uup* may suggest that Uup plays a fundamental role in limiting replisome from stalling under normal growth conditions, possibly at structured DNA sites such as inverted repeats.

The Δ*rdgB* mutant suggests another underappreciated aspect of cell division (Figs 4C and EV1A). RdgB is an enzyme that reduces the levels of non-canonical purines deoxyinosine (dITP) and deoxyxanthosine (dXTP) to prevent DNA damage associated with their incorporation into the chromosome; *rdgB* is essential for viability in a *recA⁻* background (Lukas & Kuzminov, 2006; Budke & Kuzminov, 2010). The high frequency of cell filamentation among Δ*rdgB* cells, despite the presence of functional recombination machinery, underscores the importance of a tight control of dITP and dXTP levels in the cell.

The two remaining mutants in island 22 were strains deleted for the cryptic prophage genes *croE* and *ydaS* (Figs 4C and EV1). They illustrate how this screen can identify functions for genes that are not expressed under normal growth conditions. Genes in the Keio collection were deleted by an in-frame replacement of a kanamycin-resistance cassette that has a constitutive promoter and no transcriptional terminator to ensure expression of downstream genes in operons (Baba *et al*, 2006). However, for repressed or poorly expressed operons, the kanamycin cassette promoter can lead to unregulated expression of downstream genes in operons. This was the case for the *croE* and *ydaS* deletion strains, as cells became normal in length when the kanamycin cassette was excised (Fig EV1B and C). These results, together with the normal $CV_L$ and <L> scores associated with the deletions of the downstream genes (Fig EV1B and C, Dataset EV2), suggest that it was not the loss of *croE* and *ydaS* but rather the expression of the

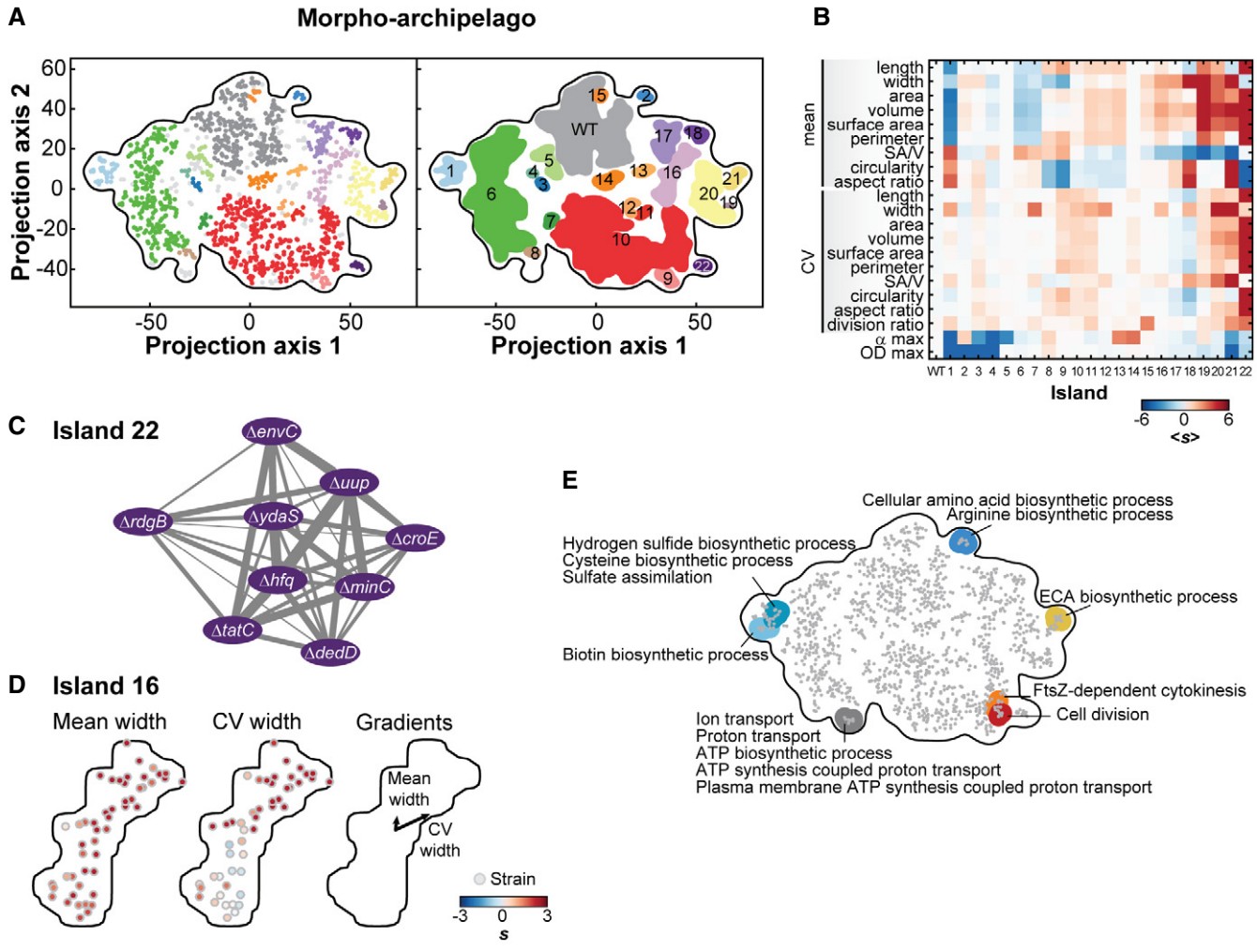

**Figure 4. The morpho-archipelago.**

A   Representative 2D t-SNE map of the 985 strains with at least one morphological or growth feature with a |s| > 3, plus the 240 independent WT replicates used as controls. Color-coded islands resulting from the dbscan algorithm (ε = 4.9, minPoints = 3) were defined by groups of strains clustering together with the same dbscan parameters in more than 90% of the generated t-SNE maps. Dots in the scatter plot (left graph) represent strains color-coded based on their island affiliation (right graph). Light gray dots represent strains that were not consistently (less than 90% of the time) associated with one of the islands.

B   Heatmap showing, for each island, the average score of each morphological and growth feature used for the construction of the map.

C   Network representation of island 22 grouping filamentous mutants. The weights were directly derived from the average distances between corresponding mutant strains in the 2D t-SNE maps.

D   Internal structure of the mean and CV of width in island 16. The average gradients of phenotypes over the area of the island 16 are represented with arrows.

E   Shown are the enriched "biological process" GO terms associated with false discovery rate below 0.05. For a list of all enriched GO terms, see Appendix Table S2.

prophage genes located directly downstream (*ymfL* and *ydaT*, respectively) that was responsible for the observed cell filamentation phenotype. Consistent with our hypothesis, it has been postulated that *ymfL* is involved in cell division (Mehta *et al*, 2004; Wang *et al*, 2010; Burke *et al*, 2013). While *ymfL* probably encodes a cell division inhibitor, the prophage gene *ydaT* likely inhibits cell division indirectly by acting on DNA replication or segregation, given the absence of well-segregated DAPI-stained nucleoids in filamentous Δ*ydaS* cells still carrying the kanamycin cassette (Fig EV1C).

Note that each island represented a continuum of phenotypes dominated by the features that lead to their clustering into one common island. Beyond the global segmentation of the morpho-space,

each island displayed some internal structure. This is illustrated in Fig 4D, which shows the gradient of the dominating (<W>) and secondary (CV$_W$) features within island 16. This fine internal organization reflects the objective function of the t-SNE algorithm, which seeks to minimize distances between similar phenoprints. This property provided us with an excellent layout to consider t-SNE maps as networks (e.g., Fig 4C), from which we could perform local functional enrichment analyses based on gene ontology (GO) terms. This approach enabled the functional annotation of the t-SNE networks while taking into account the map topology without explicit clustering (see Materials and Methods). This analysis highlighted both expected and surprising functional associations with specific morphological phenoprints (Fig 4E). For

example, the phenoprint dominated by a small cell size and growth defects (island 1), which is a hallmark of starved cells, was not surprisingly associated with an enrichment of strains deleted for genes involved in sulfate assimilation and biotin metabolism (Fig 4E).

Another example is the enrichment of genes in the enterobacterial common antigen (ECA) biosynthesis pathway (Fig 4E) among gene deletions that dramatically affected cell width control (island 21). ECA mutants tended to be wider and often lost their rod shape to form rounder cells, as shown by their high aspect ratio score (Fig EV2A, Dataset EV2). This phenotype is reminiscent of the cell shape defects caused by drugs (e.g., fosfomycin) that inhibit peptidoglycan synthesis (Kahan *et al*, 1974; Marquardt *et al*, 1994). Island 21 included other cell envelope mutants with a similar phenotype, such as gene deletions related to colanic acid (CA) biosynthesis (Dataset EV2). These results are consistent with recent studies showing that misregulation of cell shape can be caused by a competition between the ECA, CA, and peptidoglycan precursor pathways for the same undecaprenyl phosphate lipid carrier (Jorgenson & Young, 2016; Jorgenson *et al*, 2016). The cell width phenotype of gene deletions in the neighboring island 20 could be rationalized with a similar competition argument, as several of them are related to central metabolism (Dataset EV2). The metabolic genes may be essential for the production of key metabolites important for the synthesis of cell envelope precursors. The Δ*rapZ* strain, which had a severe cell width phenotype (Fig EV2B, island 20), may be an example. RapZ post-transcriptionally regulates the amount of GlmS (Gopel *et al*, 2013), which catalyzes the first committed step away from the upper glycolysis pathway and toward the synthesis of a central precursor (UDP-N-acetyl-α-D-glucosamine) for the biogenesis of peptidoglycan and ECA.

We also identified pathways associated with phenotypes that were not easy to rationalize. Deletion of genes encoding the high-affinity phosphate transporter subunits (PstA, PstC, and PstS), the associated histidine kinase (PhoR), and the adaptor protein (PhoU) led to a thin phenotype (Fig EV2C), without significantly slowing down growth (Dataset EV2). The absence of a growth defect is expected, as the growth medium is rich in phosphate that can be taken up by the low-affinity phosphate transporters. Thus, the cell width reduction cannot be associated with phosphate starvation.

Deletion of several genes encoding subunits of ATP synthase, which results in a metabolic switch to fermentation, led to a decrease in average cell width (Fig EV2C). Cultures of these deletion strains did not grow slower ($s > 0$ for $\alpha_{max}$) than WT (Dataset EV2). Furthermore, they were imaged at an $OD_{imaging}$ at least 3 times smaller than their $OD_{max}$, indicating the cell width phenotype could not be linked to the inability of some of these strains to grow to high cell density ($s < -3$ for $OD_{max}$). This result suggests that either the ATP synthase itself or differences in metabolism alter cell shape and size independently of growth rate.

Another surprise was the lack of clustering, and therefore the absence of association, among mutants expected to affect fatty acid metabolism; instead, fatty acid mutants displayed a variety of phenotypes (Dataset EV2). Previous work has shown that a reduction in fatty acid synthesis through drug treatment or deletion of the fatty acid biosynthetic gene *fabH* results in a thinner and shorter cell phenotype (Yao *et al*, 2012). Conversely, an excess of fatty acids through overexpression of the regulator *fadR* or by addition of

exogenous fatty acids leads to wider and longer cells (Vadia *et al*, 2017). These results have led to a simple model in which the amount of fatty acids and, by extension, the level of lipid synthesis determine cell size. Our data suggest a potentially more complex relationship between phospholipids and cell morphology. This is illustrated by the Δ*fadR* and Δ*fabF* strains, which were thinner ($s < -8$), but also longer ($s > 3.5$), than the parental strain (Fig EV2D). Remarkably, the width and length defects were compensatory such that the Δ*fadR* and Δ*fabF* mutants retained a normal cell area (Fig EV2D). FadR is a bifunctional transcriptional factor that activates fatty acid synthesis and represses β-oxidation, while FabF is a fatty acid chain elongation enzyme. Based on their metabolic profiles, both Δ*fadR* and Δ*fabF* strains have significant changes in the levels of phospholipids containing saturated versus unsaturated acyl chains (Fig EV2D; Garwin *et al*, 1980; Nunn *et al*, 1983; Fuhrer *et al*, 2017). These results suggest that an altered phospholipid composition, such as changes in the degree of fatty acid saturation, may be another important factor that determines the dimensions of the cell.

## Identification of genes affecting nucleoid separation and cell constriction dynamics

We applied the same t-SNE analysis to the seven cell cycle and growth features of the 397 strains displaying a severe defect ($|s| \geq 3$) for at least one cell cycle or growth feature. The 240 independent wild-type replicates were included in the analysis as controls. We robustly identified a WT island and 12 distinct mutant islands in this cell cycle space (Fig 5A). Each island was characterized by an average phenoprint (Fig 5B). Islands 6 and 7 were phenotypically close to WT. Islands 3 and 10 grouped mutants with growth defects and little to no cell cycle phenotypes (Fig 5B and C). The neighboring islands 1, 2, 9, and 11 were dominated by cell growth features with some combination of nucleoid separation and cell constriction defects. Three islands—islands 4, 8, and 12—grouped interesting gene deletion strains with cell cycle phenotypes and no significant growth defects (Fig 5B and C).

Functional analysis on all strains identified GO term enrichments with phenoprints that show strong growth phenotypes (Fig 5D). We did not find any GO term enrichment associated with cell cycle phenotypes independently of growth defects. Furthermore, the proportion of genes of unknown function (so-called "y-genes") was the highest (~30%) for cell cycle-specific islands 8 and 12 (Fig 5E). These observations highlight the limited extent of our knowledge about the genetic basis of nucleoid and cell constriction dynamics, compared to cell growth.

Our analysis of nucleoid separation and cell constriction provided a genomewide perspective on the processes affecting DNA segregation and cell division. While each event has been investigated for years at the molecular level, we know little about their coordination. We found that nucleoid separation is tightly correlated with the initiation of cell constriction across the ~4,000 deletion strains ($\rho = 0.65$, 95% confidence interval [0.63, 0.66], Fig 6A) and at the single-cell level (Appendix Fig S1H). A well-known genetic factor involved in this coordination is MatP (Mercier *et al*, 2008). This DNA-binding protein organizes and connects the chromosomal terminal macrodomain (*ter*) to the

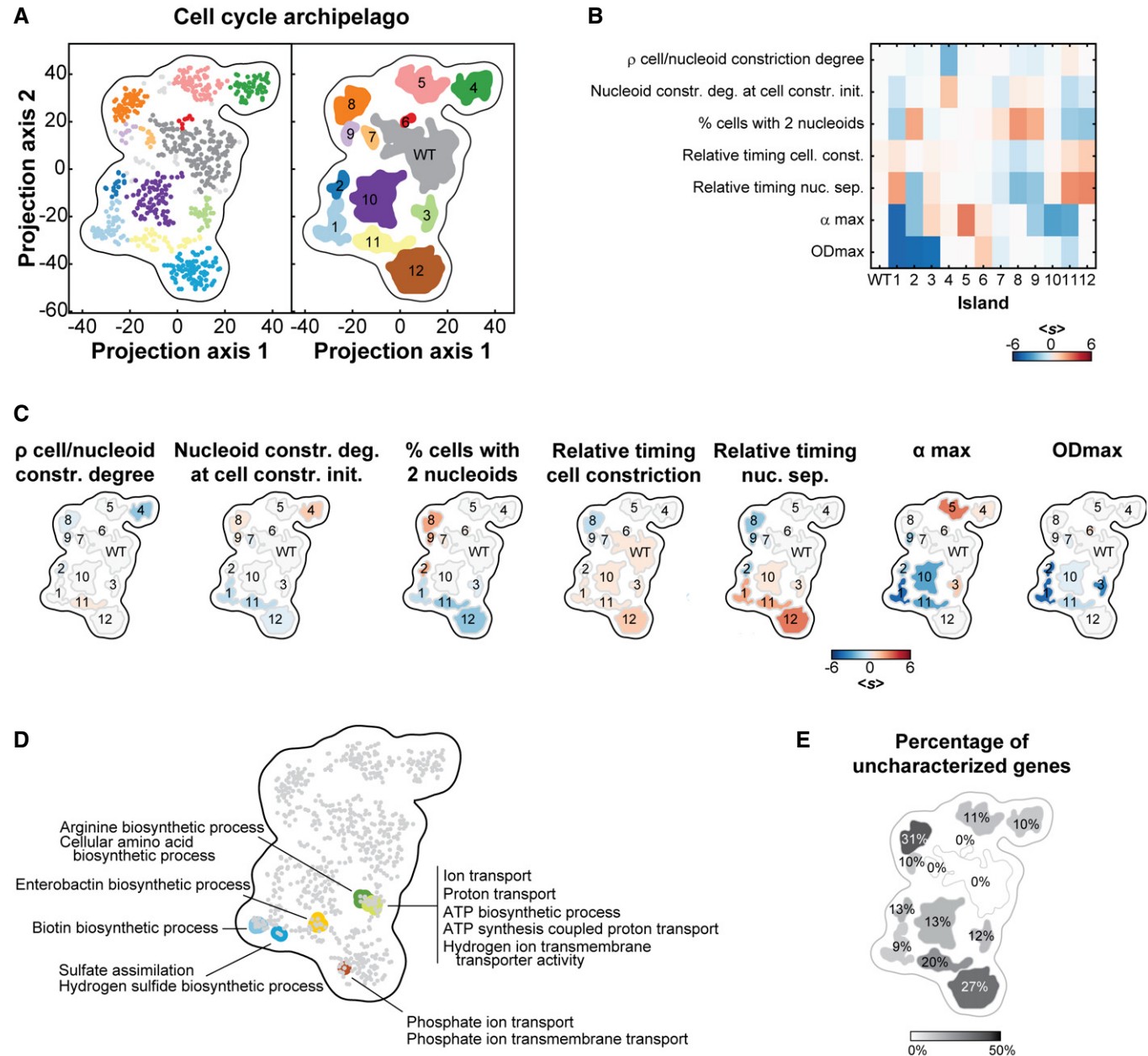

**Figure 5.  The cell cycle archipelago.**

A   Stable islands in the cell cycle archipelago. The cell cycle and growth phenoprints were used to map the 397 mutant strains with at least one cell cycle or growth feature with a |s| > 3, as well as the 240 independent WT replicates, in 2D using t-SNE. As for the morpho-archipelago, the data were clustered using the dbscan algorithm. The groups of strains clustering (dbscan parameters ε = 4.28, minPoints = 3) together in more than 90% of the maps defined an island. Dots in the scatter plot on the left represent strains and are colored with the same color code as for the island on the right graph. Light gray dots represent strains that were not consistently (less than 90% of the time) associated with one of the islands.

B   Heatmap showing, for each island, the average score of each cell cycle and growth feature used for the construction of the map.

C   Islands shown in panel (A) were colored according to their average score for each listed feature.

D   Shown are the enriched "biological process" GO terms associated with false discovery rate below 0.05. For a list of all enriched GO terms, see Appendix Table S2.

E   The islands in the cell cycle archipelago were colored according to the proportion of "y-genes" (genes of unknown function).

division machinery (Espeli *et al*, 2012). Consistent with this function, we observed that the Δ*matP* mutant, which segregated into island 8, failed to coordinate nucleoid separation with cell constriction and separated its nucleoid early while dividing at about the same cell age as WT (Fig 6A–C). The early separation of

nucleoids is in agreement with the early segregation of sister loci within the *ter* region in the Δ*matP* mutant (Mercier *et al*, 2008; Espeli *et al*, 2012) and with the proposed role of MatP in linking DNA segregation to cell division (Mannik & Bailey, 2015). The remaining 47 strains from island 8, which also displayed an early

nucleoid separation phenotype (Fig 6B), were deleted for genes that had either uncharacterized functions or functions unrelated to nucleoid dynamics (Dataset EV2). Deletion mutants that clustered closely to the $\Delta matP$ mutant within island 8 displayed a WT-like cell constriction profile as well (Fig 6B). Examples include mutants that lack the lysophospholipase L2, PldB, the DNA repair polymerase, PolA, or the poorly characterized protein, YfjK (Fig 6B and D). YfjK is a predicted helicase that has been associated with sensitivity to ionizing radiations (Byrne *et al*, 2014). On the other side of the island, mutants were characterized by not only an early nucleoid separation, but also an early initiation of cell constriction (Fig 6A, B and E), to the point that the timing of cell constriction and nucleoid separation was virtually the same. This is illustrated with $\Delta ycjV$ and $\Delta hlsU$ (Fig 6E). YcjV is a predicted ABC transporter ATPase. HslU has two functions in the cell, one as a subunit in a protease complex with HslV, and the other as a chaperone (Seong *et al*, 2000; Slominska *et al*, 2003). Since we did not observe any significant defect in cell constriction timing for the $\Delta hslV$ mutant, the $\Delta hlsU$ phenotype is more likely linked to the chaperone activity.

### Identification of cell size control mutants

How cells achieve size homeostasis has been a long-standing question in biology. While the control mechanism at play remains under debate (Amir, 2014; Campos *et al*, 2014; Iyer-Biswas *et al*, 2014; Ho & Amir, 2015; Taheri-Araghi, 2015; Taheri-Araghi *et al*, 2015; Tanouchi *et al*, 2015; Harris & Theriot, 2016; Wallden *et al*, 2016), we and others have shown that under the growth conditions considered in this study, *E. coli* follows an adder principle in which cells grow a constant length ($\Delta L$) before dividing (Campos *et al*, 2014; Taheri-Araghi *et al*, 2015). We sought to use this screen to survey the role of genes in cell length control. We first explored the relationship between mean length (<L>) and length variability ($CV_L$) among mutants. On average, short mutants ($s \leq -3$, $n = 68$) had a normal $CV_L$ (*P*-value = 0.98), while long mutants ($s \geq 3$, $n = 106$) displayed a greater $CV_L$ (*P*-value = $8.18 \times 10^{-11}$, Fig 7A).

The observation that short mutants displayed, on average, a normal $CV_L$ indicates that most of them regulate their length distribution as precisely as WT. These results suggest that the adder principle, and therefore the relative timing of cell division, is just as precise in short mutants as in WT cells. This result is interesting because short mutants have traditionally received a lot of attention in cell size control studies. A well-known short mutant in *E. coli* is the *ftsA** strain (WM1659), which is thought to misregulate size control by triggering division prematurely (Geissler *et al*, 2007; Hill *et al*, 2012). However, when we imaged the *ftsA** mutant ($n = 2,198$ WM1659 cells), we found that, similar to the trend shown by short mutants in our screen, *ftsA** cells constrict at a similar cell age as WT (Fig 7B). In hindsight, this result makes sense since the WT and *ftsA** strains have a similar doubling time (72.3 ± 2.2 min versus 69.0 ± 3.2 min, mean ± standard deviation, $n = 4$), consistent with Geissler *et al* (2007), and therefore take the same amount of time to divide. Perhaps a more appropriate way to consider short mutants with normal $CV_L$ is not as mutants that have a premature division, but as "small-adder" mutants since they add an abnormally small cell length increment $\Delta L$ between divisions.

Long mutants, on the other hand, tended to lose their ability to maintain a narrow size distribution, as $CV_L$ increased with <L> (Fig 7A). The origin for an increase in $CV_L$ may signify a loss of precision in the timing of division, but it may alternatively originate from an aberrant positioning of the division site (or both). The $\Delta minC$ mutant is an example of aberrantly large $CV_L$ (Fig 7C) due to the mispositioning of the division site, and not due to a defective adder (Campos *et al*, 2014). This class of mutants can easily be identified in our dataset by their large variability in division ratios ($CV_{DR}$). Conversely, a high $CV_L$ associated with a normal variability in division ratios points to a mutant that has a more variable $\Delta L$ between divisions.

We suspected that interesting cell size control mutants might be missed by only considering $CV_L$. The distribution of cell lengths in a population is a convolution of cell length distributions at specific cell cycle stages. Since there is significant overlap in length distributions between cell cycle stages, a substantial change in $CV_L$ at a specific cell cycle stage (e.g., cell constriction) does not necessarily translate into obvious changes in $CV_L$ of the whole population, as shown in simulations (Fig EV3). Our screen allowed us to identify constricting cells and hence to determine the cell length variability for the stage of cell constriction. This cell cycle stage-specific analysis identified $\Delta mraZ$ as a potential gain-of-function cell size homeostasis mutant (Fig 7C). For this mutant, division ($CV_{DR}$, Dataset EV2) and growth rate (Eraso *et al*, 2014) were normal, but the length distribution of its constricted cells ($CV_L = 0.05$) was remarkably narrower than that of WT constricted cells ($CV_L = 0.12$). MraZ is a highly conserved transcriptional regulator that downregulates the expression of the *dcw* cluster (Eraso *et al*, 2014), which includes cell wall synthesis and cell division genes (Ayala *et al*, 1994). Our data suggest that MraZ and the regulation of the *dcw* cluster affect the balance between cell growth and division.

### Dependencies between cellular dimensions and cell cycle progression

A fundamental question in biology is how cells integrate cellular processes. A common approach to address this question is to look at covariation between processes or phenotypes following a perturbation (e.g., mutation, drug treatment). However, perturbations that affect the same system (i.e., perturbing a single process or pathway) can lead to misinterpretation, as the perturbation may abolish a given dependency between two features or may affect the co-varying phenotypes independently. Increasing the number of independent perturbations has two major effects. First, it alleviates the interpretation problem by averaging out the specific effect associated with each perturbation (Sachs *et al*, 2005; Collinet *et al*, 2010; Liberali *et al*, 2015). The Keio collection consists of mutants affected in a wide variety of cellular processes, allowing us to examine whether specific features correlate across many different genetic perturbations. Second, the large number of genetic perturbations increases our confidence in any calculated correlation (or lack thereof) between features. To illustrate, let's consider two independent features (i.e., true correlation = 0). Using an analytical solution (Fisher, 1915, 1921), one can show that the calculated 95% confidence interval (CI) for the Pearson correlation is between −0.63 and +0.63 if the sample size is 10. The calculated 95% CI is very wide because two uncorrelated features can easily appear positively or

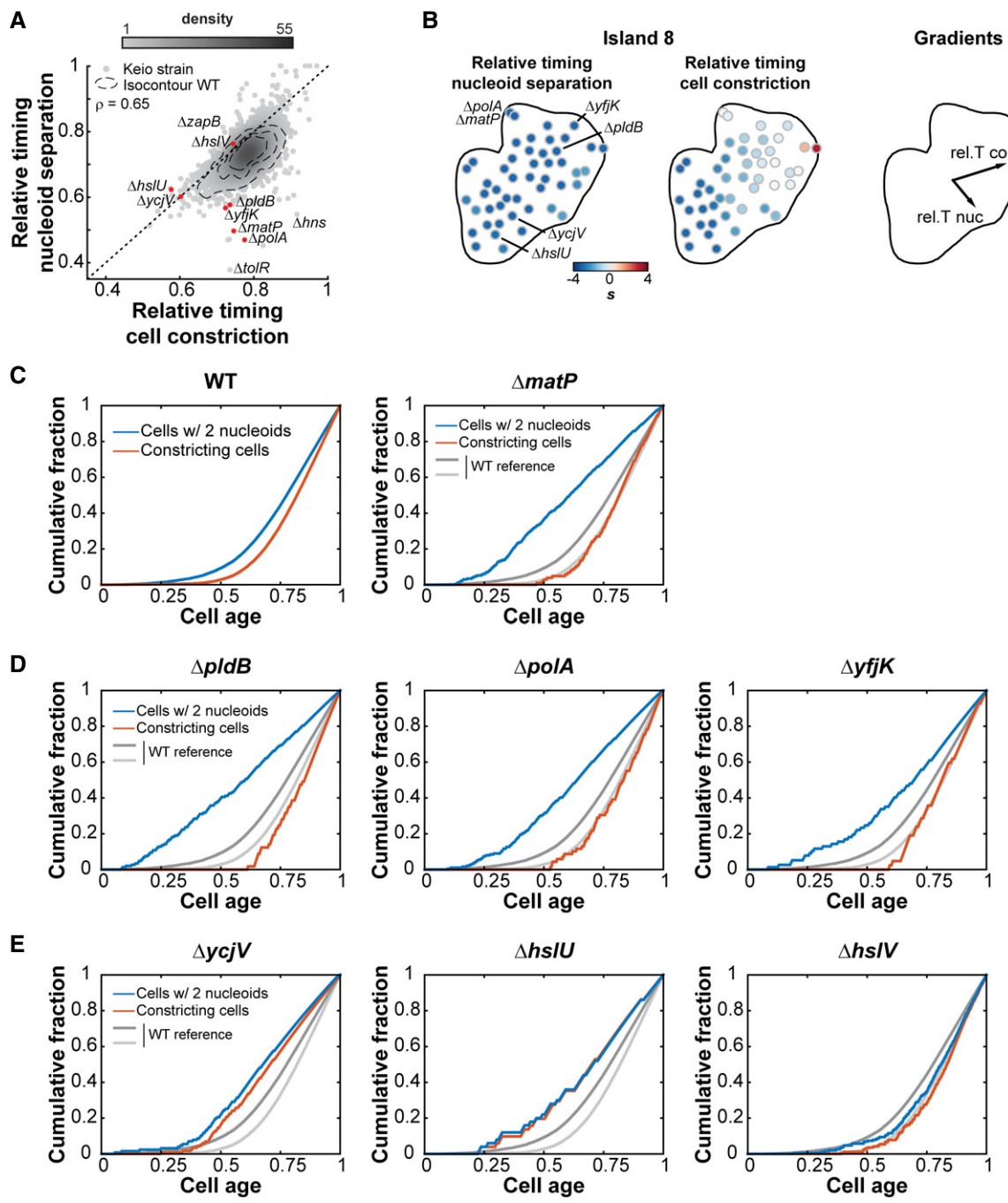

**Figure 6. Nucleoid separation and cell constriction dynamics.**

A   Scatter plot of the relative timing of cell constriction versus the relative timing of nucleoid separation. The gray scale indicates the density of dots in a given area of the chart. The dotted contours represent the 0.5, 0.75, and 0.95 probability contours of the 240 WT replicates. The Pearson correlation is ρ = 0.65, with a 95% CI = [0.63, 0.66]. The black-dotted diagonal represents the line where a strain should be if both nucleoid separation and cell constriction happen at the same time. Red dots highlight strains shown in panels (C, D, and E).

B   Close-up view of the cell cycle island 8 with each dot representing a Keio strain colored according to the two features driving the clustering of the 48 strains. The relative timing of nucleoid separation is the dominant feature of island 8, while the relative timing of cell constriction drives the layout of the strains within the island.

C   Average dynamics of nucleoid separation and cell constriction for WT and the Δ*matP* mutant strain. The cumulative distributions of the fraction of cells with two nucleoids (blue) and of the fraction of cells with a constriction degree above 0.15 (red) were plotted against cell age. Cell age was calculated according to the rank of each cell based on their cell length with the formula $age_i(F) = -\ln(1-F/2) / \ln(2)$, where $F$ represents the fraction of cells with a cell length equal or below the length of cell $i$ (Wold *et al*, 1994).

D   Same plots as in (C) for three strains clustering in island 8 with the Δ*matP* strain. The WT curves shown in (C) were plotted in gray for comparison.

E   Same plots as in (C) for two island-8 strains, Δ*ycjV* and Δ*hslU*, and the Δ*hslV* strain, which does not partition in island 8.

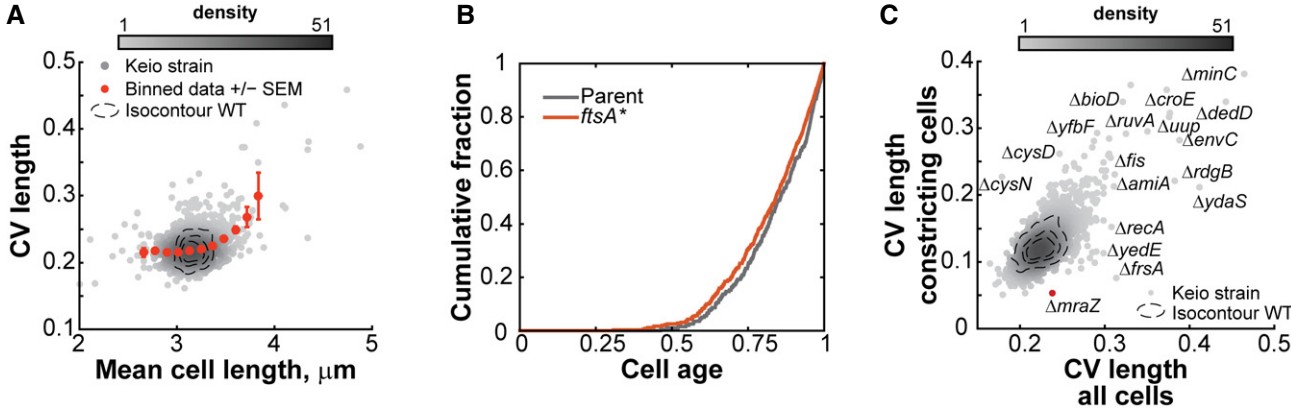

**Figure 7. Cell length regulation mutants.**

A   Scatter plot of the mean cell length versus the CV of the length for all the strains. The gray color levels indicate the density of points in the vicinity of each strain. The orange dots and error bars represent the mean and standard error of the mean per bin.

B   The cumulative distribution of the proportion of constricting cells for the *ftsA\** mutant and its parent were plotted against cell age. We measured the degree of constriction of all cells using Oufti. For each strain, we had two independently acquired image sets ($n = 496$ and $1,168$ cells for the WT replicates and $n = 692$ and $1,506$ cells for the *ftsA\** replicates). The distributions of the degree of constriction were not significantly different among the four datasets at a threshold *P*-value of 0.01 using a Kruskal–Wallis multicomparison test. Moreover, Bonferroni-corrected *post hoc* pairwise tests did not allow a distinction between WT and *ftsA\** samples.

C   Scatter plot of the CV of the cell length for the whole population versus the CV of the cell length for constricting cells only. The contour lines represent the 0.5, 075, and 0.95 probability envelopes of the 240 independent WT replicates. The gray color levels indicate the density of points in the vicinity of each strain. The Δ*mraZ* strain discussed in the text is highlighted in red.

negatively correlated if the sample size is small. The 95% CI shrinks down to [−0.06, +0.06] if the sample size is 1,000. Therefore, the large number and variety of mutants in our study provided an opportunity to identify global effects and dependencies between morphological, cell cycle, and growth phenotypes through correlation analysis.

To build an interaction network, we used the information-theoretic algorithm ARACNE (Margolin *et al*, 2006). This method considers all pairwise correlations between features at the same time and identifies the most relevant connections by removing those that are weak or that can be explained via more correlated paths. In this analysis, we focused on 10 quantitative non-collinear features that describe cellular dimensions (<L>, <W>, <A>), nucleoid size (<NA>), population growth ($\alpha_{max}$, $OD_{max}$), nucleoid separation (rel. T nuc), cell constriction (rel. T const), and their interdependency ($\rho$CD, $CDN_{C0}$; see Materials and Methods). The resulting network recovered obvious connections, such as the relation of cell area with cell length and width. It also showed that growth rate features displayed virtually no connectivity to cell size or cell cycle features (Fig 8A). This was further illustrated by the close-to-zero correlations between growth rate and any of the 24 morphological and cell cycle features considered in our screen (Fig EV4A).

Another interesting lack of connection was between <L> and <W> (Fig 8A), as these two variables were largely uncorrelated (Fig 8B). With $n = 4,227$, our estimated Pearson correlation ($\rho = 0.06$) is associated with a narrow 95% CI between 0.03 and 0.09. By subsampling this large dataset, we can show how a decrease in sample size increases the likelihood of obtaining erroneous positive or negative correlations (Fig EV5). The lack of correlation between <L> and <W> is interesting from a cell size control standpoint. If *E. coli* was controlling its size by sensing its volume, surface area, or the ratio between the two, as *Caulobacter crescentus* does (Harris *et al*, 2014), we would expect a global anti-correlation between length and width

such that an increase in cell length would be, on average, compensated by a decrease in width, and vice versa. The lack of correlation argues that cell length and width are controlled independently in *E. coli*, at least under our growth conditions.

Some features, however, displayed strong covariation across the 4,000 genetic perturbations. For instance, the mean cell area and mean nucleoid area (considering the sum of nucleoids in the cell) were highly positively correlated ($\rho = 0.84$, 95% CI [0.83, 0.85]), in a growth rate-independent manner (Fig 8C). In wild-type cells, nucleoid size linearly increases with cell size throughout the cell cycle (Junier *et al*, 2014; Paintdakhi *et al*, 2016). Here, we found that nucleoid size scales with cell size across ~4,000 mutants despite their effects on different cellular functions: Small mutants had a small nucleoid size, and big mutants had a big nucleoid size (Fig 8C). This remarkable linear relationship held true regardless of the number of nucleoids per cell (Fig 8D). In addition to its strong positive correlation with the average cell size, the average nucleoid size was negatively correlated with the relative timing of nucleoid separation ($\rho = -0.48$, 95% CI [−0.50, −0.45], Fig 8E). These connections suggest a dependency between size and cell cycle features: the bigger the cell is, the bigger the nucleoid is (Fig 8C), and the earlier nucleoid separation and cell constriction occur in relative cell cycle units (Fig 8E and F). This dependency is highlighted by the overall structure of the interaction network (Fig 8A), which reveals that the cell cycle features (yellow nodes) are primarily connected to the cellular dimension features (blue nodes) through the dimensions of the nucleoid (gray node).

# Discussion

In this study, we used a multi-parametric approach to quantitatively survey the role of the non-essential *E. coli* genome in cell shape, cell

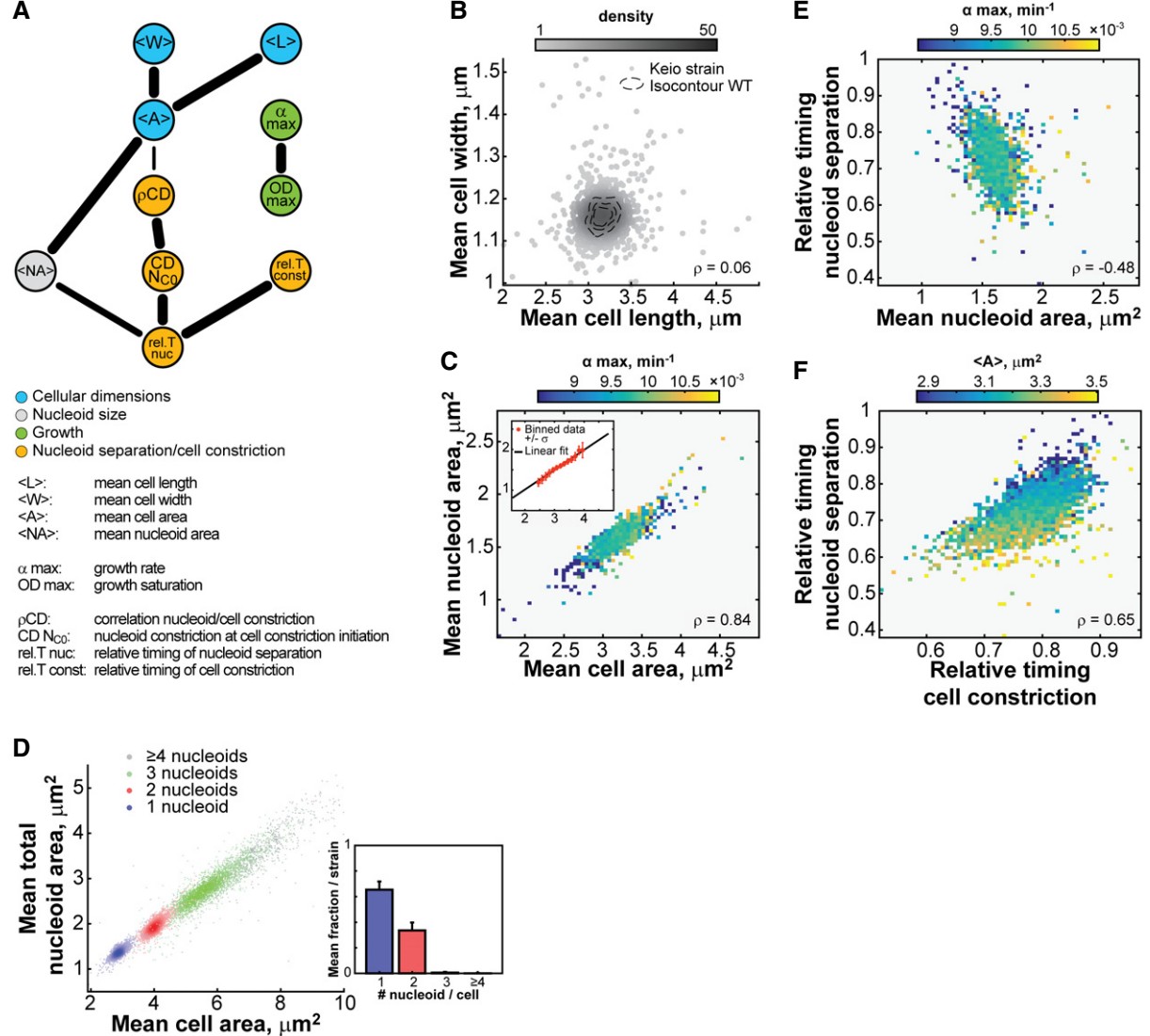

**Figure 8.    Interdependence of cell morphogenesis and cell cycle progression.**

A   Network showing the functional relationship between 10 non-collinear morphological, growth, and cell cycle features. The network is an undirected network highlighting the most informative connections detected by the ARACNE algorithm. The thickness of an edge represents the fraction of the networks containing this specific edge after bootstrapping the network 200 times, from 70% (thinnest) to 100% (thickest).

B   Scatter plot of the normalized mean cell length and mean cell width of all 4,227 Keio strains and 240 WT replicates. Each dot represents a strain, and the gray level illustrates the density of neighbors in the vicinity of each point in the graph. The dotted contours represent the 0.5, 0.75, and 0.95 probability envelopes of the 240 WT replicates. The correlation between mean cell width and mean cell length is low ($\rho$ = 0.06, 95% CI [0.03, 0.09]).

C   Heatmap showing the mean growth rate value for data binned by both mean cell area and mean nucleoid area. The cell and nucleoid areas are strongly correlated ($\rho$ = 0.84, 95% CI [0.83, 0.85]). The median value of $\alpha_{max}$ per bin is color-coded according to the color scale. The inset highlights the linearity of the relationship between mean cell area and mean nucleoid area. The orange dots and error bars represent the binned data, with the black line showing the best linear fit to the binned data.

D   Scatter plot of the mean cell area versus the mean nucleoid area for cells with 1, 2, 3, or ≥ 4 nucleoids for each strain. The histogram in the inset illustrates the average proportions of cells with 1, 2, 3, or ≥ 4 nucleoids per strain. Although there were typically few cells in each strain with 3 or ≥ 4 nucleoids, at least one cell with ≥ 3 nucleoids was detected for 61% of the strains.

E   Heatmap showing the mean growth rate value for data binned by both the mean nucleoid area and the relative timing of nucleoid separation. The mean nucleoid area negatively correlates with the relative timing of nucleoid separation ($\rho$ = −0.48, 95% CI [−0.50, −0.45]).

F   Heatmap showing the mean cell area value for data binned by both the relative timing of cell constriction and the relative timing of nucleoid separation. The relative timing of cell constriction is strongly correlated to the relative timing of nucleoid separation ($\rho$ = 0.65, 95% CI [0.63, 0.66]).

size, cell growth, and two late cell cycle stages, nucleoid separation and cell constriction. The results provide a valuable resource of phenotypic references for both characterized and uncharacterized genes, as well as a rich dataset to explore the correlation structure between cellular dimensions, growth, and cell cycle features at the system level.

The large proportion of genes and the wide variety of functions impacting cell size and shape and the progression of late cell cycle stages (Figs 2 and 3, Appendix Fig S5) underscore the degree of integration of cell morphogenesis and cell cycle progression in all aspects of *E. coli* cell physiology. It also implies that most morphological and cell cycle phenotypes cannot easily be imputed to a specific pathway or cluster of genes. In fact, genes involved in the same cellular process can have very different, and even sometimes opposing, effects. Genes associated with translation illustrate this concept. Deletion of ribosomal subunit genes leads to a diversity of morphological phenotypes, such as thin (Δ*rplY*), wide (Δ*rpsO*), short (Δ*rpsT*), and short and thin (Δ*rpsF*; Dataset EV2). This diversity of phenotypes is also observable for deletions of genes encoding enzymes that modify ribosomal RNAs or tRNAs (e.g., Δ*rsmD* cells are long, whereas Δ*rluD* and Δ*truA* cells are wide, and Δ*mnmC* cells are long, wide, and variable in size). The latter suggests an unexpected role for RNA modifications in cell morphogenesis.

Overall, this study greatly expands the number of genes associated with cell morphogenesis (874) and the cell cycle (231). Notably, it provides a phenotype for 283 genes of uncharacterized function (out of 1,306 y-genes). The Keio collection, as any genome-wide deletion collection (Teng *et al*, 2013), is no stranger to gene duplications and compensatory mutations (Yamamoto *et al*, 2009; Otsuka *et al*, 2015). Although their occurrence can impact interpretation at the gene level, "suppressor" mutants tend to display more WT-like behavior. It is, therefore, possible that we have underestimated the actual number of mutants displaying altered phenotypic characteristics or underrated the severity of the phenotype of a given deletion. Importantly, "incorrect" mutants have no effect on our global correlation analyses, as the latter does not rely on the genetic identity of the mutants.

Our study reveals new phenotypes for previously characterized gene deletions. We have already mentioned above the unexpected filamentation phenotype of the Δ*uup* strain (Figs 4C and EV1A) and have proposed a tentative connection between Uup's known function (precise transposon excision) and DNA damage through replisome stalling. We also identified unanticipated links. For example, the requirement for lysophospholipase L2 (PldB) in the coupling of nucleoid separation and cell constriction (Fig 6D) suggests a connection between phospholipid metabolism and the coordination of late cell cycle stages. Previous works have also linked phospholipid metabolism to cell morphology (Yao *et al*, 2012), showing that fatty acid availability dictates the capacity of the cell envelope to expand, ultimately affecting cell size (Vadia *et al*, 2017). We found that mutants with an altered degree of saturated versus unsaturated phospholipids have an abnormal length and width (Fig EV2D). It is, therefore, tempting to speculate that not only the amount, but also the composition of fatty acids, plays an important role in cell shape and size control. Phospholipid composition determines the chemical and physical properties of the cell membrane (Dowhan & Bogdanov, 2002), which is known to affect the function of cell division and morphogenesis proteins such as MinD and MreB (Mileykovskaya *et al*, 2003; Kawazura *et al*, 2017).

By combining the t-SNE and dbscan algorithms, we were able to cluster strains with similar phenoprints into islands (Figs 4 and 5). This granular representation of the phenotypic space allowed us to expand on well-studied archetypal phenotypes such as "filamentous" and "fat" (islands 22 and 21 of the morpho-archipelago, respectively,

see Fig 4). This classification also allowed us to populate less well-studied phenotypes, from which we can gain new insight into cell morphogenesis and the cell cycle. For example, the substantial number of thin mutants reported here may prove as valuable as fat mutants to study cell morphogenesis from a different angle. The clustering results also revealed entirely new classes of mutants. In that respect, islands 4, 8, and 12 of the cell cycle archipelago are particularly interesting because they offer a genetic toolkit to explore nucleoid and cell constriction dynamics, which have remained poorly understood despite their essential role in cellular replication.

It is important to note that the phenoprints reported in this study are tied to the specific experimental conditions of the screen. Cell size is well known to vary with nutrient conditions (Schaechter *et al*, 1958), indicating that the behavior of the Keio mutants may be different in other environments. Differences in growth conditions also lead to different metabolic requirements and growth limitations. For instance, none of the mutant strains auxotrophic for nucleotides were able to grow in our synthetic medium, which lacks nucleotide precursors. We also note that growth in 96-well plates likely corresponds to micro-aerophilic conditions. Accordingly, we identified morphological deviations for strains deleted for genes known to be only expressed under micro-aerophilic or anaerobic conditions, revealing new metabolic connections to cell morphogenesis. For example, deletion of *ybcF*, which is predicted to encode an enzyme involved in anaerobic purine degradation (Smith *et al*, 2012), results in a fat cell phenotype (Dataset EV2).

In this study, each gene deletion can be seen as a perturbation. The sheer number of perturbations (~4,000) guarantees a large number of independent perturbations and offers a unique opportunity to infer the underlying correlation structure between the different phenotypes (Fig 8A). Such relationships, or lack thereof, can be very informative. For instance, we found that growth rate is not predictive of cell size. This is an interesting finding because current theoretical models of cell size control generally include growth rate as a variable. Recently, the original growth law was modified to include a second variable, the C + D period (time of DNA replication + time between the end of DNA replication and cell division), that influences cell size (Zheng *et al*, 2016; Si *et al*, 2017). Although we do not have specific measurements of C + D periods and thus cannot directly compare our results to this general growth law, the observed lack of correlation between growth rate and cell size remains surprising, especially for the fast-growing mutants (Appendix Fig S7). Growth inhibition and nutrient limitation experiments have shown that the C + D period increases with slower growth rates (Kubitschek & Newman, 1978; Wallden *et al*, 2016; Si *et al*, 2017). It is therefore possible that a lengthening of the C + D period in slow-growing mutants compensates for the growth rate difference, masking the relationship between the growth rate and cell size in our dataset. However, under our nutrient-growth conditions of overlapping DNA replication cycles (Appendix Fig S1A), the C + D period is supposed to have reached a plateau, i.e., a minimal value that remains constant at even faster growth rates (Cooper & Helmstetter, 1968; Helmstetter, 1968; Wallden *et al*, 2016; Si *et al*, 2017). Therefore, we would not expect fast-growing mutants to have a shorter C + D period that compensates for their faster growth rate. Future studies on these fast-growing mutants could be enlightening.

The relative timings of nucleoid separation and cell constriction are independent of growth rate (Fig EV4B and C). The absence of

correlation between growth rate and the relative timing of these cell cycle events was also observed for the wild-type strain when the growth rate was varied by changing the composition of the growth medium (Den Blaauwen *et al*, 1999). Collectively, our findings show that the cell can accommodate a large range of sizes and relative timings of nucleoid segregation and cell division with no effect on growth rate, and vice versa. This flexibility may offer greater evolvability of cellular dimensions and cell cycle progression.

The complexity of cellular systems can sometimes be reduced to simple quantitative relationships, which have been very useful in identifying the governing principles by which cells integrate various processes (Scott & Hwa, 2011). Our correlation analysis identified a strong linear relationship between nucleoid size and cell size. This remarkable scaling property is independent of growth rate and holds across the wide range of cellular perturbations present in the ~4,000 deletion strains examined in this study (Fig 8C and D). This result draws a striking parallel with the 100-year-old observation that nucleus size scales with cell size in eukaryotes (Conklin, 1912), an empirical relationship that has been reported for many eukaryotic cell types since then (Vukovic *et al*, 2016). This suggests a universal size relationship between DNA-containing organelles and the cell across taxonomic kingdoms, even for organisms that lack a nuclear envelope.

Our information-theoretic Bayesian network analysis (Fig 8) enabled us to go beyond pairwise correlations by integrating the complex set of interdependencies between cell morphogenesis, growth, and cell cycle events. This analysis unveiled an unexpected connection between average cell size and the relative timings of nucleoid separation and cell constriction through nucleoid size across thousands of genetic perturbations (Fig 8E and F). This finding suggests that the size of the nucleoid is an important element of the coordination mechanism between cell morphogenesis and the cell cycle.

# Materials and Methods

### Bacterial strains and growth conditions

The Keio collection contains 3,787 annotated single-gene in-frame deletion strains, 412 strains (referred to as JW strains) with kanamycin cassette inserted at unknown locations, and the remainder (28) were repeats (Baba *et al*, 2006).

All strains, including *E. coli* K12 BW25113 (Datsenko & Wanner, 2000) and derivatives (strains from the Keio collection), as well as *E. coli* K12 MG1655 and isogenic *ftsA*\* WM1659 strains (Geissler *et al*, 2007), were grown in M9 medium (6 g/l $Na_2HPO_4 \cdot 7H_2O$, 3 g/l $KH_2PO_4$, 0.5 g/l NaCl, 1 g $NH_4Cl$, 2 mM $MgSO_4$, 1 μg/ml thiamine) with 0.2% glucose as the carbon source and supplemented with 0.1% casamino acids.

### Screening setup and microscopy

All *E. coli* strains were grown overnight at 30°C in 96-well plates in M9 supplemented with 0.1% casamino acids, 0.2% glucose, and kanamycin (30 μg/ml). Cultures were diluted 1:300 in 150 μl of fresh M9 medium supplemented with 0.1% casamino acids and 0.2% glucose and grown in 96-well plates at 30°C with continuous

shaking in a BioTek plate reader. DAPI was added to the cultures to a final concentration of 1 μg/ml 15–20 min prior imaging. All (parent and mutant) strains were sampled within a very narrow range of $OD_{600\ nm}$ (0.2 ± 0.1; min = 0.108; max = 0.350) corresponding to the exponential growth phase (Appendix Fig S1D). We did not detect any trend between morphological/cell cycle features and the $OD_{600\ nm}$ at which each culture was sampled. Cells were deposited (0.5 μl per strain) on a large, 0.75-μm-thick, M9-supplemented agarose pads with a multichannel pipette. The pads were made by pouring warm agarose containing supplemented M9 medium between a (10.16 × 12.7 × 0.12 cm) glass slide and a (9.53 × 11.43 cm) n° 2 cover glass (Brain Research Laboratories, Newton, MA, USA).

Microscopy was performed on an Eclipse Ti-E microscope (Nikon, Tokyo, Japan) equipped with Perfect Focus System (Nikon) and an Orca-R2 camera (Hamamatsu Photonics, Hamamatsu City, Japan) and a phase-contrast objective Plan Apochromat 100×/1.45 numerical aperture (Nikon). The initial field of view for each strain was chosen manually, and nine images were taken automatically over a 3 × 3 square lattice with 200-nm step, using 80-ms exposure for phase-contrast and 600-ms exposure for the DAPI channel using Nikon Elements (Nikon).

### Image processing

Cell outlines were detected using the MicrobeTracker software (Sliusarenko *et al*, 2011). All data processing was then performed using MATLAB (The MathWorks Inc., Natick, MA, 2000). Custom-built codes were used to automate the aggregation of data from the cell outlines of all the strains. Data are available in Datasets EV1 and EV2.

For cell and nucleoid detections, we consistently used the same parameters (See Appendix for parameters). In order to avoid unnecessary bias in the cell outlines, the parameters defining the initial guess for the cell contour fit were set to intermediate values, while the parameters constraining the fit of the final outline were set to negligible values. For example, we increased the *fsmooth* parameter value to 100 in order to capture both short and long cells, and we set the width spring constant parameter *wspringconst* to 0 so as to avoid biasing the cell width estimate toward the initial guess value. The edges in the DAPI fluorescence signal were detected with Oufti's objectDetection tool (Paintdakhi *et al*, 2016) which is based on a Laplacian of Gaussian filtering method that takes into account the dispersion of the point spread function (PSF) of our microscopy setup at a wavelength of 460 nm (input parameter $\sigma_{PSF}$ set to 1.62 pixels).

### Rifampicin run-out experiments

The number of ongoing replication cycles was examined in run-out experiments (Skarstad & Katayama, 2013). BW25113 cells were grown at 30°C either in M9 glycerol or M9 medium supplemented with 0.1% casamino acids, 0.2% glucose, and 1 μg/ml thiamine (as for the Keio screen described here). Cells were grown up to exponential phase and then treated for 3 h with 30 μg/ml cephalexin and 300 μg/ml of rifampicin prior to overnight fixation in 70% ethanol at 4°C. Cells were washed twice with phosphate-buffered saline and then stained with DAPI (1 μg/ml) prior to imaging on a

PBS-containing agarose pad. In M9 glycerol medium, *E. coli* BW25113 cells do not start a new round of replication before the previous one ended (Cooper & Helmstetter, 1968; Wang *et al*, 2011). This growth condition was used as a control to estimate the DAPI intensity corresponding to 1 and 2 genome equivalents.

### Data analysis

*Dataset curation—Support Vector Machine (SVM) model*
Due to the size of the dataset (> 1,500,000 cells detected globally), we adopted an automated approach to identify poorly (or wrongly) detected cells across the entire dataset. We developed an SVM model based on 16 normalized features: cell length, cell width, cell area, cell volume, cell perimeter, cell constriction degree, division ratio, integrated phase signal, integrated DAPI fluorescence signal, mean cell contour intensity in phase contrast, variability of cell width along the cell, nucleoid area, single-cell nucleoid variability, cell circularity ($2*\pi*$cell area/(cell perimeter)$^2$), nucleoid intensity, and number of nucleoids. We trained a binary classifier (positive or negative) over wild-type strain replicates as well as 419 mutants with the most severe morphological defects prior to data curation. We visually scored 145,911 cells and used 30% of them (43,774) to train the model. The model was evaluated using a k-fold cross-validation approach, leading to a generalized misclassification rate of 10%. We used the remaining 70% of the data set (102,137 cells) to validate the model. This SVM classifier achieves a balanced classification rate of 84% and features an AUROC of 0.94 (Appendix Fig S1E). Furthermore, the resulting group of false negatives was not significantly different from the true positives (Appendix Fig S1F and G), indicating that the classification did not introduce a bias by excluding a specific class of "good" cells from the analysis.

*Data processing*
For each feature, we checked and corrected for any bias associated with plate-to-plate variability, differences in position on the 96-well plates, timing of imaging, and optical density of the culture (Appendix Figs S2–S4). For each plate, we set the median values of each feature, *F*, to the median feature value of the parental strain.

The *F* values were transformed into normalized scores by a transformation akin to a *z*-score transformation but more robust to outliers.

$$s = 1.35 \times \left( F_i - median\left( F_i^{WT} \right) \right) / iqr\left( F_i^{WT} \right),$$

where $F_i$ is the corrected value for the mutant strains for feature *i*, $F_i^{WT}$ is the value for the wild-type strain for feature *i*, and iqr stands for interquartile range. As the interquartile range of normally distributed data is equal to 1.35 times their standard deviation, we scaled the score by this factor so as to express the scores in terms of standard deviations away from the median.

The temporal biases for the fraction of cells committed (or not) to division and the fractions of cells with 1, 2, or more nucleoids were corrected using a Dirichlet regression to maintain the relative proportions between classes (Appendix Fig S3; Maier, 2014). In an exponentially expanding population of growing *E. coli* cells at steady state, the fraction of cells in the population before the occurrence of a cell cycle event is related to the cell age at which this event occurs by a monotonic relationship (Collins & Richmond,

1962). The proportions of cells at different cell cycle stages are therefore used to infer the relative proportions of different cell cycle stages and the cell age at which a specific cell cycle event occurs (Powell, 1956; Collins & Richmond, 1962). These inferences only rely on the assumption that the population of cells is in a steady state leading to a stable distribution of cell ages (see the appendix of Wold *et al*, 1994 for a detailed, mathematical description). One limitation of this approach is that only relative timings or proportions can be inferred, and no conclusions should be drawn on the absolute duration of the different periods without any other hypothesis. For instance, a reduction of the fraction of cells with one nucleoid could result either from an actual reduction in the cell cycle period associated with one nucleoid or from a lengthening of the period associated with cells with two or more nucleoids. In both cases, we can, however, conclude that the separation of nucleoids happens earlier in relative cell cycle units. Another limitation is that only population-level average timings can be obtained. Although these average relative timings would also be obtained when averaging the behavior of many individual cells of that same population, we cannot quantify single-cell level variability in these timings (and its potential cross-talk with morphological features of individual cells).

*Data exploration, dimensionality reduction, and clustering*
A similarity measure between strains was needed to identify and separate different phenoprints. Pearson correlations or Euclidean distances classically provide such similarity measures, and principal component analysis (PCA) and/or hierarchical or k-means clustering are often used. However, PCA tends to explode datasets and Pearson correlations do not always reflect the desired type of similarity. As an extreme example, consider two strains with two phenoprints that are proportional, one with values within a very small score range, such as [−1 1], while the other with score values spanning the [−10 10] range. These two strains will get a maximal similarity measure through a correlation analysis, despite the fact that the first strain is wild-type-like while the other is an outlier. Instead, we chose to use a recently described algorithm, called t-distributed stochastic neighbor embedding, or t-SNE (van der Maaten & Hinton, 2008), to project our multidimensional datasets in two dimensions and generate, at the same time, similarity measures between strains. t-SNE estimates low-dimensional space distances between points based on their similarity, as opposed to dissimilarity as in the case of PCA, thereby highlighting local similarities rather than global disparities.

We used the stochastic nature of the t-SNE algorithm to evaluate the robustness of the resulting projection by repeating the procedure multiple times (*n* = 100 for each t-SNE map). We coupled this dimensional reduction procedure with a density-based clustering algorithm, dbscan (Ester *et al*, 1996), to group strains with similar phenoprints. The two input parameters of the dbscan algorithm, ε and minPoints, were optimized so as to generate a maximum number of islands without separating the bulk of WT strains in two or more islands. Islands include strains that clustered together more than 90% of the time.

The convergence of the dimensionality reduction was verified by repeating the t-SNE dimensionality reduction on subsamples of the initial dataset (1,225 × 21 matrix). In this approach akin to cross-validation, we generated 50 partitions of either 1,200 or 1,201 strains, holding out disjoint sets of points, with the

cvpartition built-in function in MATLAB, and repeated the dimensional reduction with the t-SNE algorithm 10 times for each partition. We first compared the pairwise distances between the points in each of these 500 t-SNE maps with the pairwise distances between the corresponding points in the t-SNE map presented in Fig 4A. The distribution of Pearson correlation coefficients between these sets of distances, calculated with a kernel density estimation function (Botev *et al*, 2010), is illustrated in Appendix Fig S8A (red curve). This distribution is highly similar to the distribution obtained from the repetition of the t-SNE dimensional reduction on the full dataset (1,225 strains—blue curve in Appendix Fig S8A) and suggests that the algorithm converges toward a global minimum. The bimodality in the distribution of the Pearson correlation coefficients is not due to specific subsamples (Appendix Fig S8B) and rather reflects the stochasticity of the t-SNE algorithm and the low weight carried by large distances in the map. For example, the displacement of a well-isolated island relative to others may not impact strongly the minimized score of this t-SNE map, but would definitely reduce the Pearson correlation coefficient between this map and the reference t-SNE map presented in Fig 4A. Using the same parameters as for the full dataset ($\varepsilon = 4.9$, minPoints = 3), we verified that the dbscan algorithm results in a similar clustering as in the global map. The resulting clusters for each t-SNE map associated with a subsample of the dataset were compared with the clustering output presented in Fig 4A and B. For each reference cluster (islands in Fig 4A), we identified all the representative points of this cluster in the subsample t-SNE map and calculated the ratio between the maximal number of these points clustering together in this subsample map divided by the total number of representative points of the reference cluster in this subsampling. This ratio is akin to the Jaccard index for each reference cluster, and all indexes for a given map were averaged to provide a score to each t-SNE map. The distribution of these scores (10 scores per subsample corresponding to the 10 t-SNE maps calculated for each subsample) is represented in the boxplot in Appendix Fig S8C. The average indexes are typically above 0.9, which reflects the threshold used to generate the clusters as points clustering in more than 90% of the t-SNE maps. The same reproducibility of clustering in the subsample t-SNE maps is also a good indication that the t-SNE dimensional reduction converges toward an optimal embedding.

## Map exploration

Each t-SNE map is a similarity map, and can therefore be treated as a network where the nodes represent strains and the edges the Euclidean distance between strains in the t-SNE map. Building up on recent quantitative network analysis tools (Baryshnikova, 2016), we calculated the local enrichment in the maps of different strain-associated attributes, such as COG and GO terms. Briefly, the sum of the attributes in a local area (within a radius around each point, defined as the 1-percentile of the distribution of all the pairwise distances between points) was compared to a background score (defined as the average score obtained over 1,000 identical maps with randomly permutated attributes) with a hypergeometric test. The significant local enrichments were considered at a threshold of 0.05 after adjusting for false discovery rate correction with the Benjamini–Yekutieli (BY) procedure,

taking into account dependencies between tests (Benjamini & Yekutieli, 2001). The SAFE algorithm proposed by Baryshnikova (2016) was implemented as a MATLAB function mapEnrich.m (see Code EV1).

## Cluster of orthologous gene enrichment analysis

We associated *E. coli* BW25113 genes with COGs using the web server (Van Domselaar *et al*, 2005). The enrichment analyses were performed using a custom-built algorithm in MATLAB based on a two-tailed hypergeometric test to compute *P*-values, which were subsequently adjusted with the Benjamini–Hochberg (BH) false discovery rate procedure (Benjamini & Hochberg, 1995) (Code EV2). Because the COG categories are largely independent, we did not consider any correction for the dependence between tests.

## Gene ontology analysis

We used ontologies from the Gene Ontology website (http://www.geneontology.org/ontology/gene_ontology.obo, version 2016-05-27; Ashburner *et al*, 2000), and annotations were obtained from EcoCyc for *E. coli* strain MG1655 (Keseler *et al*, 2013). Analysis was performed using a MATLAB custom-built algorithm that includes a hypergeometric test to compute *P*-values that were subsequently adjusted with the BY false discovery rate procedure (Benjamini & Yekutieli, 2001; Code EV2).

## Bayesian network

The Bayesian network presented in Fig 8 was generated in R with the bnlearn package (Scutari, 2010), using the ARACNE algorithm as described in Margolin *et al* (2006). The network was bootstrapped 200 times, and all the edges were identified in more than 70% of the networks. We assessed the strength and the origin of collinearity among features using Belsley diagnostic method (Belsley *et al*, 1980), with the built-in collintest.m function in MATLAB. We excluded features associated with a "condition number" above the classical threshold of 30.

## Calculation of confidence intervals

As the true correlation between two normally distributed variables approaches 1 or −1, the probability density distribution of the estimated correlation becomes highly skewed and far from normal. For Pearson correlations ($\rho$), this distribution can be transformed using Fisher z-transform to approximate a normal distribution for any true correlation value *r* (Fisher, 1915):

$$z = \frac{1}{2}\ln\left(\frac{1+r}{1-r}\right).$$

The variable *z* follows a normal distribution with mean $\frac{1}{2}\ln\left(\frac{1+\rho}{1-\rho}\right)$ and standard error $\frac{1}{\sqrt{n-3}}$, where $\rho$ and $n$ are the estimated Pearson correlation value and the sample size, respectively.

Using this transformation, the limits of the distribution covering the 95% most probable values of the distribution can be calculated using the 2.5 and 97.5 percentiles of the normal distribution centered on $\frac{1}{2}\ln\left(\frac{1+\rho}{1-\rho}\right)$ with standard deviation $\frac{1}{\sqrt{n-3}}$.

$$[z_{low}, z_{up}] = z \pm z_{97.5\%}\sqrt{\frac{1}{n-3}}.$$

The inverse transformation on these confidence boundaries on $z$ is then used to calculate the 95% confidence upper and lower limits for the estimated correlation value ρ.

$$\rho_{low} = \frac{e^{2z_{low}} - 1}{e^{2z_{low}} + 1}; \quad \rho_{up} = \frac{e^{2z_{up}} - 1}{e^{2z_{up}} + 1}$$

Unfortunately, a similar approach cannot be employed for an estimated Kendall correlation (τ) value (Long & Cliff, 1997). To generate 95% CIs, we used a bootstrapping approach (Efron & Tibshirani, 1993). Correlation values were calculated between 5,000 resampled (with replacement) variable pairs. The 2.5 and 97.5 percentiles of the obtained distribution of correlation values were subsequently taken as the respective lower and upper boundaries of the bootstrapped 95% CIs.

### Data representation

All graphs were generated using MATLAB, except for the networks in Figs 4C and 8A panels, which were created using Cytoscape v3.2 (Shannon *et al*, 2003) and the Rgraphviz package in R (Hansen *et al*, 2017), respectively. For Fig 4C, we used the built-in, edge-weighted, spring-embedded algorithm in Cytoscape. We considered the pairwise Euclidean distances between the nine strains of island 22 as the weights of the edges connecting the nodes (or strains).

The density scales in scatter plots represent the number of points around each point in a radius equal to the 0.03 percentile of the pairwise distances distribution.

The WT isocontours representing the 0.5, 0.75, and 0.95 probability envelopes for the 240 WT replicates were calculated using a 2D kernel density estimation function over a 128-by-128 lattice covering the entire set of points. The bandwidth of the kernel was internally determined (Botev *et al*, 2010).

### Simulations of cell length distributions

Cell length distributions at any given cell age were assumed to be log-normally distributed with different dispersion values. The CV of the distribution for the WT strain (CV = 0.11) was previously experimentally determined (Campos *et al*, 2014). The cell length distributions at 100 different ages equidistantly distributed between 0 (birth) and 1 (division) were convolved with the cell age distribution, assuming an exponentially growing culture, $\mathrm{Pr}(age) = 2^{-age}$.

### Data and software availability

The imaging data from this publication have been deposited to the BioStudies database (https://www.ebi.ac.uk/biostudies/). The accession number for our image dataset is S-BSST151. The dataset files containing the raw data, normalized data, and scores for all the strains are available as Datasets EV1 and EV2. The computer codes are available as ZIP files Code EV1 and EV2.

**Expanded View** for this article is available online.

### Acknowledgements
We are grateful to the Yale *E. coli* Genetic Stock Center for providing a large number of strains. We also thank Pr. William Margolin for the kind gift of the *E. coli* MG1655 strain and the *ftsA** derivative. This work was partly supported by the National Institutes of Health (R01 GM065835 to C.J.-W.). We also thank the Jacobs-Wagner laboratory for fruitful discussions and for critical reading of the manuscript. M.C. was partly funded by a fellowship from the "Fondation pour la Recherche Médicale" (ARF20160936199). C.J.-W. is an investigator of the Howard Hughes Medical Institute.

### Author contributions
CJ-W and MC designed experiments. GSD, SKG, II, and MC performed experiments. MC performed high-throughput imaging and statistical analyses. CJ-W supervised the project. CJ-W, MC, FC, II, and SKG wrote the manuscript.

### Conflict of interest
The authors declare that they have no conflict of interest.

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
