## [Review Process File · Molecular Systems Biology]

Genome-wide phenotypic analysis of growth, cell morphogenesis and cell cycle events in *Escherichia coli*

Authors: Manuel Campos, Sander K. Govers, Irnov Irnov, Genevieve S. Dobihal, François Cornet and Christine Jacobs-Wagner

Review timeline:

Submission date:	7 th February 2017
Editorial Decision:	15 th March 2017
Revision received:	4 th April 2018
Editorial Decision:	17 th May 2018
Revision received:	28 th May 2018
Accepted:	29 th May 2018

Editor: Maria Polychronidou

Transaction Report:

1st Editorial Decision

15th March 2017

Thank you again for submitting your work to Molecular Systems Biology. We have now heard back from the three referees who agreed to evaluate your manuscript. As you will see below, the reviewers appreciate that the presented data could be a useful resource for the field. They raise however a series of concerns, which should be carefully addressed in a major revision.

I think that the referees' recommendations are rather clear, so there is no need to repeat the points listed below. Please feel free to contact me in case you would like to discuss any specific point in further detail.

REFeree REPORTS

Reviewer #1:

Campos et al. describes *E. coli* cell morphologies and nucleoid staining in a previously established knockout library. The approach is high-throughput imaging of on agarose pads and automated image analysis similar to what has been done previously by Kuwada et al Mol Micro 2015 and Peters et al Cell 2016. The data presented in the supplementary Excel files is however novel, impressive and possibly useful.

Concerns

There is no time lapse imaging of the stains which is limiting considering that most interesting part of the conclusions deals with cell cycle properties. The paper would be significantly strengthened by actual time lapse imaging over generations. Alternatively, it should be clearly stated that the cell cycle progression is indirectly estimated from constrictions and nucleoid localizations in snap-shots and the possible misinterpretations resulting from not actually following individual cells from division to division need to be addressed.

Treating the knock-outs as perturbations that does not change the wiring of the system seems risky. For example, it is not necessarily so that the length to width ratio is regulated normally in a knock-out that is defect in setting the width. In this respect the gene knock-out perturbation is radically different than altering the growth conditions without changing the hardware. I would like to see a much better justification for the knock-out library approach for describing the E. coli cell cycle regulation.

The interpretation corresponding to the black line in fig 7A seems to be a stretch. To me the data seems essentially uncorrelated. If I am not wrong, the increased CV for long cells only applies to <1% of the strains.

The statement that lack of correlation between L and W in the mutants (page 16:416-422) implies that the cells do not add a constant volume per generation is interesting. Do the authors imply that the cells may still add a constant length per generation (length-adder model) independent of the width of that strain? If so, it would be interesting to see a time lapse adder experiments, of the type that the group has done before, with some different width mutants.

Reviewer #2:

Campos et al conduct a genome-wide phenotypic screen in E. coli, by systematically scoring the key morphological parameters in the Keio collection. Then, the authors analyze the data in different ways, and put particular emphasis on a) addressing the relationship of cell size and different aspects of growth rate control, and b) touch on the potential of phenomics to add new annotation to so far uncharacterized genes. This is a highly useful study, and I'm very much in favor of its publication. There would be the potential in the one or the other place to make it more useful for the community though.

General comment: This study stands or fails with the quality of the imaging as recorded and its data analysis. I feel comfortable with the data analysis, I'm however not an expert on HTP imaging techniques, and cannot judge how reasonable the thresholds have been defined, and how well the false positive/negative rates perform in comparison to other studies in this field. While all reads good, its important that at least one of reviewer is able to comment on this - as its such an important aspect of the study.

Specific comments.

I personally thank the authors for addressing the misleading believe that cell size is causally determined by growth rate, which is clearly not the case (but a growth rate dependency on all quantitative phenotypes is almost imprinted in the field). The reason is perhaps, that in some other situations (not when genes are deleted, but when cells grow under different nutritional conditions) there are correlations between growth rate and cell size, and these wrongfully implied causality. I was wondering if the authors my want to expand the discussion a little about this... especially as they are clearly not the first ones in question this relationship, but this data is fresh and strong to address the problem in knock-out strains.

Growth conditions: Obviously, morphological phenotypes are sensitive to the growth conditions, in this case standard lab conditions are sued (i.e. temperature, glucose, amino acid supplement). Certainly, the authors cannot make such a huge screen under many conditions to address the variability exhaustively, but it is surprising the authors do not even mention it. Is it possible on the basis of existing data, or on the basis of i.e. re-profiling a small subset of the strains under different nutritional conditions, to provide a rough estimate how much phenotypes are likely missed? Apparently, the amino acid supplements will suppress a huge number of growth phenotypes that

arise from defects in amino acid biosynthetic metabolism.

Secondary and compensatory mutations in the Keio collection. Fair enough, they exist, and this is biology. But I think their confounding impact should be discussed. Do the authors perhaps have some examples which would allow to derive some quantities? If not they should simply discuss this problem with Reference to previous studies in this or other species (i.e. there is good yeast data out there about the problem).

Finally, I miss a comparison with other functional genomic screens in the Keio collection. I am aware this might be some work, but this indeed would make the study so much more valuable. The phenotypic clusters must associate to other clusters as determined in other laboratories screens, and perhaps explain some of the derived phoneme clustering mechanistically. This would also give the study a bit more the touch of a mechanistic rather than a purely descriptive study.

The way the datasets and the results (i.e. phenotypic clusters) are made available and made easily searchable for the community, could be improved.

Reviewer #3:

The manuscript from Campos et al. describes an image-based screen for the *E. coli* single-gene knockout library, in which they quantified cellular morphologies, growth, and cell cycle-related properties, and then connected the quantitative phenotypical observations with functional analysis to associate the phenotypes with gene functionality. They additionally studied the correlation across genetic knockouts for cell dimensions, growth, and nucleoid dynamics, which revealed some systematic relationship across those metrics.

Overall, the manuscript is clearly written and the experimental data provide good resources on single-cell phenotypes across genetic knockouts. However, this manuscript presents system-scale data without extracting general rules or validating their discoveries with specific examples, especially for Figures 2 to 5. I would like to see some more concrete biological conclusions from all the statistics presented. In addition, I believe there are several issues on both experimental designs and data interpretation that need to be addressed.

1. A general comment: all the experiments in this manuscript were performed in M9 media with casamino acids and glucose at 30 {degree sign}C. In such relative poor media with low temperature, cells grow slower and may be under quite different physiological states compared to other experiments where cells grown in LB, and/or experiments performed at 37 {degree sign}C. For instance, multi-fork replication may not occur in the current condition, which means that cells may regulate DNA replication quite differently. Therefore, unless the authors can provide evidences that their discoveries still hold true for fast-growing conditions, it would be good to clearly state the differences in experimental conditions for all the conclusions made in the manuscript.

2. Due to the strong dependence of cellular morphology to experimental conditions, it is very critical to compare all the strains under the same conditions. One concern is that since different strains have different maximum OD, then the OD at which imaging is performed (~0.2) does not necessarily guarantee exponential growth, especially for those with low maximum OD. So, additional controls are needed for these strains to confirm that morphology and cell cycle features are still independent of OD at which imaging is performed.

3. Also, relating to the above issue about experimental conditions. Presumably imaging 48 strains on a large agar pad takes tens of minutes, during which cells would grow on the pad. Such growth may alter cell dimensions, and introduce newly synthesized unlabeled DNA. Thus, the bias introduced by imaging sequence should be addressed before further analysis.

4. In Fig 7A, there are many strains that have WT-like lengths but larger length CVs, which also indicates a mis-regulation in cell size homeostasis. Have the authors looked at any strains in that region?

5. Lines 416-421: the authors made a strong statement that since no correlation in length/width was observed, cells are not controlling their sizes by monitoring volume or surface area. However, since many of the genetic knockouts can directly affect volume growth and/or surface area growth (e.g. if

cell wall incorporation or PG precursor synthesis is affected, cells are likely to reduce surface area to compensate; a similar effect is reported by Harris and Theriot, Cell 2016), then it is almost expected that volume or surface area cannot be conserved across genetic backgrounds. Therefore, the authors don't have enough evidence to conclude that mean length and width are independently regulated.

Some minor points:

1. In Fig 2, the authors make the reference by using a Gaussian distribution. Would it make more sense to use the distribution of wild-type controls?
2. In Fig 3, category H is highly enriched for morphological features with scores ≤ 3 . Do the authors have any comments on that?
3. Fig 7B, p value is needed for the comparison.

We thank the reviewers for their feedback and thoughtful comments. We have addressed their concerns and suggestions; please see below for a point-by-point response (in blue). We would also like to acknowledge that during the revision, we noticed a mistake in our dataset. We incorrectly used a non-final dataset in which a small subset of the strain data had not been normalized, slightly affecting the scores of these few strains in the original submission. We sincerely apologize for this oversight. While this mistake only affected a small fraction of the data, we nevertheless repeated all the analyses and remade all the figures using the correct dataset. As a result, some numbers and figures are a bit different. However, the conclusions have remained the same. We have also simplified the interaction network in Figure 8 to focus on the important quantitative features, which, we believe, will help the reader visualize the core message.

Response to reviewer's comments

Reviewer #1:

Campos et al. describes E. coli cell morphologies and nucleoid staining in a previously established knockout library. The approach is high-throughput imaging of on agarose pads and automated image analysis similar to what has been done previously by Kuwada et al Mol Micro 2015 and Peters et al Cell 2016. The data presented in the supplementary Excel files is however novel, impressive and possibly useful.

We thank the reviewer for his/her recognition of the novelty and potential impact of our work. We agree that the imaging approach we used has been described before. However, we would like to point out that there are several important technical aspects in our approach that are novel compared to Kuwada et al, Mol Microbiol (2015), Peters et al, Cell (2016), and others.

- 1) *Support Vector Machine (SVM) curation of automated cell detection.* No current automated cell detection program is perfect; all create poor cell outlines on occasion. Manual detection of these poor cell contours is not practical for genome-wide studies. As a result, poor cell contours are not curated. To our knowledge, we are the first to curate our dataset of aberrant cell contours by constructing, validating and implementing a supervised machine learning model. We believe that this approach will be useful for many other future imaging studies.
- 2) *Combination of tSNE and dbscan algorithms for the classification and clustering of high-dimensional datasets.* The Kuwada et al (2015) and Peters et al (2016) used principal component analysis (PCA) to analyze their large dataset. PCA is a great method for identifying outliers. However, it cannot identify groups of mutants with similar phenotypic properties. To demonstrate this point, we have added a PCA analysis of our data in the revised manuscript (Fig. S6). To address this issue, we have developed a new approach that combines tSNE and dbscan algorithms, which allowed us to transform large, high-dimensional datasets into discrete clusters of mutant strains that share a similar phenoprint (combination of phenotypic features). We also believe that this approach will be useful for other future imaging studies.
- 3) *Information-theoretic Bayesian network analysis using the ARACNE algorithm to identify global effects and dependencies between morphological, cell cycle and growth phenotypes.* We also describe a novel implementation of this ARACNE algorithm for network construction to go beyond simple pairwise correlations and integrate the complex set of phenotypic variables.

Concerns

There is no time lapse imaging of the stains which is limiting considering that most interesting part of the conclusions deals with cell cycle properties. The paper would be significantly strengthened by actual time lapse imaging over generations. Alternatively, it should be clearly stated that the cell cycle progression is indirectly estimated from constrictions and nucleoid localizations in snap-shots and the possible misinterpretations resulting from not actually following individual cells from division to division need to be addressed.

Inferring the average relative timing of cell cycle events from a snapshot of a population of cells (e.g., FACS, fluorescence microscopy) has been widely used in both eukaryotic and bacterial fields for decades and has been the basis for the description of cell cycle progression of many cell types, including *E. coli* cells. The inferences rely on the assumption that the population of asynchronous cells is in a steady state leading to a stable distribution of cell ages, which allows the extraction of cell cycle characteristics given a known fraction cells having passed (or not passed) a specific cell cycle stage (see the appendix of Wold et al, EMBO J 1994 for a detailed, mathematical description). A very large number of bacterial cell cycle studies rely on this assumption (Burdett et al, J Bacteriol 1986, Campbell et al, Bacteriol Rev 1957, Collins et al, J Gen Microbiol 1962, Cooper et al, JMB 1968, Donachie, Nature 1968, Flåtten et al, PLoS Genet 2015, Hill et al, PLoS Genet 2012, Koch et al, J Gen Microbiol 1982, Paulton et al, J Bacteriol 1970, Powell, J Gen Microbiol 1956, Schaechter et al, J Gen Microbiol 1958, Skarstad et al, EMBO J 1986, Vischer et al, Front Microbiol 2015, Woldringh, J Bacteriol 1976, Woldringh et al, J Bacteriol 1977), and recent studies directly comparing both types of observation (time-lapse and snap-shots) have verified the validity of this assumption (Kafri et al, Nature 2013, Mangiameli et al, PLoS Genet 2017, Trojanowski et al, mBio 2015, Grangeon et al, PNAS 2015, Reyes-Lamothe et al, Cell 2008). However, we realized that this is not common knowledge and have therefore revised the text to describe how we extract cell cycle dynamics from snapshots of populations (lines 697-712).

To further illustrate the validity of this approach and its applicability in inferring the average relative timing of cell cycle events, we analyzed previous microfluidic datasets (Campos et al, Cell 2014 and Paintdakhi et al, Mol Microbiol 2016) and directly compared the average relative timing of both cell constriction and nucleoid segregation, as calculated from time-lapse and snapshot data (randomly chosen frames from the same experiment). We found that these were indeed in near-perfect agreement (see figure below).

Figure. Comparison of methods for inferring the average relative timing of cell cycle events. A. Cell constriction profiles of individual BW25113 *E. coli* cells (n = 1528) grown in microfluidic chambers in M9 medium supplemented with 0.2 % glucose, 0.1 % casamino acids and 1 $\mu\text{g/ml}$ thiamine. Mean cell constriction at a given relative cell age is shown in white. B. Determination of the relative timing of constriction based on time-lapse and snapshot data. For time-lapse experiments, the relative cell age at which the average constriction profile exceeded the threshold value of 0.15 was taken as the relative timing of cell constriction. To generate the snapshot dataset, cell constriction values of individual cells were extracted from two arbitrarily chosen frames from the time-lapse experiment (yielding a similar number of cells as employed per strain in our genome-wide screen) and the fraction of non-constricting cells (constriction < 0.15) was used to calculate the relative timing of cell constriction. This process was repeated 10 times (n = 339-409 cells), mean and standard deviation of these repetitions are shown. C. Determination of the relative timing of nucleoid segregation based on time-lapse and snapshot data. For these experiments, BW25113 *hupA::hupA-mCherry* cells (n = 740 cells) were grown in microfluidic chambers in M9 medium supplemented with 0.2 % glucose, 0.1 % casamino acids and 1 $\mu\text{g/ml}$ thiamine. For time-lapse experiments, the relative timing of nucleoid segregation was determined by taking the average of the relative timing of nucleoid segregation for each individual cell. To generate the snapshot dataset, an identical approach as for the constriction dataset was used. Nucleoid segregation information was extracted from two arbitrarily chosen frames, and the fraction of cells containing a single nucleoid was used to calculate the relative timing of nucleoid segregation. This process was repeated 10 times (n = 213-432 cells), mean and standard deviation of these repetitions are shown. Note that the difference between the relative timing reported here and the value in the manuscript is a consequence of differences in experimental conditions (microfluidic chamber vs. 96-well plate and HU α -mCherry label vs. DAPI staining). For the time-lapse datasets, the error bars indicate the bootstrapped estimates of the standard error of the mean.

Treating the knock-outs as perturbations that does not change the wiring of the system seems risky. For example, it is not necessarily so that the length to width ratio is regulated normally in a knock-out that is defect in setting the width. In this respect the gene knock-out perturbation is radically different than altering the growth conditions without changing the hardware. I would like to see a much better justification for the knock-out library approach for describing the *E. coli* cell cycle regulation.

We agree, and we do not consider the individual deletions as perturbations that do not change the wiring of the system. However, different deletions likely affect different aspects of the wiring and by considering a large number of gene deletions, we aimed to average out specific effects associated with each individual deletion. As importantly, the KEIO collection consists of mutants that are affected in many different cellular processes, which allowed us to identify phenotypes that co-vary across a large variety of perturbations (variety of pathways and processes affected). Conversely, changing growth condition (e.g., carbon source) consists of a single type of perturbation. A single perturbation could affect two phenotypes independently. For example, a change in metabolism affects growth rate and cell size. It does not mean that growth rate affects cell size, as metabolism could affect them independently. However, if two variables strongly co-vary across many types of perturbations, causality is more likely. The greater the number of independent perturbations, the more meaningful the correlation becomes.

Our ARACNE approach also allowed us to go beyond pairwise comparison and to reveal the correlation structure of morphological, cell cycle and growth phenotypes (i.e., the wiring). A similar strategy was used in eukaryotes (Collinet et al, Nature 2010). Knowledge of global effects and dependencies suggest potential causalities within the properties of the cellular system, as recently reviewed (Liberali et al, Nat Rev Genet 2015). For example, in our analysis, we illustrate cell morphology to be linked to the cell cycle via nucleoid size. In addition to uncovering systems properties, our approach has the added benefit of identifying functions, pathways and previously uncharacterized genes involved in cell morphogenesis, population growth, nucleoid dynamics and cell division.

We agree that this was not well explained in the original manuscript. We have therefore expanded the text and added relevant citations to clarify the reasoning behind our approach (lines 420-440).

The interpretation corresponding to the black line in fig 7A seems to be a stretch. To me the data seems essentially uncorrelated. If I am not wrong, the increased CV for long cells only applies to <1% of the strains.

The bilinear fit was a guide to the eyes and has been removed in the revised manuscript to avoid confusion. Our conclusion is based on the trend shown by the binned data. The binned data for mutants with high cell length (score > 3) includes 106 mutants, which is non-negligible. Student's t-tests show that the higher average CV_L value of the long mutants (n = 106 strains) is significantly different from that of WT-like mutants (p-value = $8.18 \cdot 10^{-11}$), whereas the average CV_L values of WT-like and short mutants (score < -3, n = 68 cells) do not differ statistically (p-value = 0.98). We have added this information in the figure legend.

The statement that lack of correlation between L and W in the mutants (page 16:416-422) implies that the cells do not add a constant volume per generation is interesting. Do the authors imply that the cells may still add a constant length per generation (length-adder model) independent of the width of that strain? If so, it would be interesting to see a time lapse adder experiments, of the type that the group has done before, with some different width mutants.

We agree with the reviewer. The suggested experiments would, however, only be conclusive if we performed microfluidics on many different width mutants and analyzed not only their size expansion during the cell cycle but also their initiation mass and C+D periods given recent studies (Si *et al.*, *Curr Biol*, 2017 and Zheng *et al.*, *PNAS* 2016). These experiments would represent an enormous amount of work, considering that a new microfluidic mold would have to be constructed for each mutant (microfluidic devices are sensitive to variation in cell width; cells that are too thin are washed away, whereas cells that are too thick don't get in or get jammed). We believe that this is beyond the scope of this study, but we are interested in pursuing this type of work in the future.

Reviewer #2:

Campos et al conduct a genome-wide phenotypic screen in *E. coli*, by systematically scoring the key morphological parameters in the Keio collection. Then, the authors analyze the data in different ways, and put particular emphasis on a) addressing the relationship of cell size and different aspects of growth rate control, and b) touch on the potential of phenomics to add new annotation to so far uncharacterized genes. This is a highly useful study, and I'm very much in favor of its publication.

We thank the reviewer for his/her appreciation of our work.

There would be the potential in the one or the other place to make it more useful for the community though.

General comment: This study stands or fails with the quality of the imaging as recorded and its data analysis. I feel comfortable with the data analysis, I'm however not an expert on HTP imaging techniques, and cannot judge how reasonable the thresholds have been defined, and how well the false

positive/negative rates perform in comparison to other studies in this field. While all reads good, its important that at least one of reviewer is able to comment on this - as its such an important aspect of the study.

The reviewer is correct that threshold selection is of critical importance in a study of this kind. After correcting for temporal biases and plate-to-plate variability, we transformed the obtained feature values for the mutant strains into normalized scores that reflect the number of standard deviations the feature of interest (of a given strain) deviates from the median of the WT (240 replicates). We subsequently employed a threshold of an absolute score $|s| \geq 3$ for at least one feature. This threshold is conservative and the gold standard in high throughput analyses (Krzywinsky & Altman, Nat Methods 2013).

Specific comments.

I personally thank the authors for addressing the misleading believe that cell size is causally determined by growth rate, which is clearly not the case (but a growth rate dependency on all quantitative phenotypes is almost imprinted in the field). The reason is perhaps, that in some other situations (not when genes are deleted, but when cells grow under different nutritional conditions) there are correlations between growth rate and cell size, and these wrongfully implied causality. I was wondering if the authors my want to expand the discussion a little about this... especially as they are clearly not the first ones in question this relationship, but this data is fresh and strong to address the problem in knock-out strains.

We agree with the reviewer that the growth rate dependency of cell size is imprinted in the field. As suggested by the reviewer, we expanded the discussion two ways: 1) we have added plots to show that there is no correlation between growth rate and any of the 24 morphological and cell cycle features to draw the message home (Fig. EV4A). 2) In light of recent publications (Zheng et al, PNAS 2016 and Si et al, Curr Biol, 2017), we have also added a discussion to address how our results relate to the “generalized growth law”, which is a modified version of the “growth law” recently introduced by the Jun and Liu labs (see lines 559-577).

Growth conditions: Obviously, morphological phenotypes are sensitive to the growth conditions, in this case standard lab conditions are sued (i.e. temperature, glucose, amino acid supplement). Certainly, the authors cannot make such a huge screen under many conditions to address the variability exhaustively, but it is surprising the authors do not even mention it. Is it possible on the basis of existing data, or on the basis of i.e. re-profiling a small subset of the strains under different nutritional conditions, to provide a rough estimate how much phenotypes are likely missed? Apparently, the amino acid supplements will suppress a huge number of growth phenotypes that arise from defects in amino acid biosynthetic metabolism.

We agree that phenotypes may vary the growth conditions. We did mention in the discussion of the first submission (lines 378-385) that our data are restricted to the growth condition used in the study, but evidently not in a clear way. We have therefore revised the text to provide a clearer statement (lines 543-554).

Re-profiling a subset of strains under different growth conditions would be useful only if the subset was large enough to represent the dynamic range of values we observed for the >30 features. The confidence interval around a correlation value between two features tells us about our confidence about the level of correlation detected. We can estimate the sample size required to reach a sufficiently tight distributions

of correlation values based on our dataset (by bootstrapping). In the case of poorly correlated features such as <L> and <W>, about 400 strains would be required to estimate the correlation value ± 0.1 (95% confidence interval [-0.1, +0.1]). Furthermore, we think that such study would be particularly insightful if we include other cell cycle markers to gather more quantitative cell cycle information. We are very interested in performing such a large study in the future, but we believe that it is beyond the scope of this work.

Secondary and compensatory mutations in the Keio collection. Fair enough, they exist, and this is biology. But I think their confounding impact should be discussed. Do the authors perhaps have some examples which would allow to derive some quantities? If not they should simply discuss this problem with Reference to previous studies in this or other species (i.e. there is good yeast data out there about the problem).

It is indeed true that the KEIO collection, as any genome-wide deletion collection (Teng et al, Mol Cell 2013), is no stranger to gene duplications and compensatory mutations (Otsuka et al, NAR 2015). Although their occurrence can impact genome-wide analyses such as ours, incorrect mutants (i.e., mutants in which the native behavior is masked by a compensatory mutation or gene duplication) likely display more WT-like behavior. Consequently, it is possible that we have underestimated the actual number of mutants displaying altered phenotypic characteristics or underrated the severity of the phenotype of a given deletion. We do not have examples to derive some quantities, but we have added some text in the discussion to address the potential confounding impact of these mutations (lines 505-513).

Finally, I miss a comparison with other functional genomic screens in the Keio collection. I am aware this might be some work, but this indeed would make the study so much more valuable. The phenotypic clusters must associate to other clusters as determined in other laboratories screens, and perhaps explain some of the derived phoneme clustering mechanistically. This would also give the study a bit more the touch of a mechanistic rather than a purely descriptive study.

As rightfully noted by the reviewer in a previous comment, the phenotypes may vary with the growth conditions. The only genome-wide *E. coli* study that used the same growth medium as our study is a metabolomics study (Fuhrer et al, Molecular Systems Biology, 2017). At the reviewer's suggestion, we used data from this study to identify correlations between metabolic profiles and cellular phenotypes (mean cell length, mean cell width, mean cell area, mean nucleoid area, max. growth rate, optical density at saturation, correlation between cell and nucleoid constriction degrees, nucleoid constriction degree at the onset of cell constriction, relative timing of cell constriction and relative timing of nucleoid separation). We categorized each peak corresponding to an annotated metabolite according to the z-score reported for each peak in Fuhrer *et al.* (2017) based on an absolute threshold of 2 to use the score they consider significant in their study. These categories were used to test for enrichment among mutants with similar phenoprints. Some associations were easy to explain. For example, small and slow growing strains were associated with starvation (auxotrophy) based on their metabolic profiles (see figure below). These mutants also displayed a late timing of nucleoid separation and cell constriction, likely due to their starving condition.

We also identified more interesting associations between phenoprints and metabolic profiles, but they were far more difficult to interpret. As shown in the figure below, the phenoprint of thin cells with a late timing of nucleoid separation and cell constriction was associated with the depletion or accumulation of

certain metabolites. These mutants displayed normal growth characteristics. Therefore, the aberrant metabolic profile was not simply due to starvation.

While we are happy to share this information with the reviewers, we would prefer to exclude it from the manuscript at this stage. We are confident in our results, but not in the interpretation. We are also concerned that readers may too quickly assume causality in observed correlations that may be the result of an independent co-variation.

That said, comparison between the metabolomic data and ours led to an interesting, worth-publishing observation regarding fatty acid mutants that display an altered phospholipid profile. Recent studies have shown that reduction of fatty acid synthesis leads to a thinner and shorter cell phenotype while excess of fatty acids results in longer and wider cells (Vadia et al., *Current Biology* 2017, Yao *et al.*, *PNAS* 2012). These results have led to a simple model in which the amount of fatty acids and, by extension, the level of lipid synthesis determine cell size. Our analysis suggests a more complex relationship between phospholipids and cell morphology in which not only the amount, but also the composition (e.g., saturated vs. unsaturated) of fatty acids play a role in cell size regulation. These new results are presented in lines 301-320 and Fig EV2D.

Figure. Cross-analysis with a genome-wide metabolic profiling of the Keio collection. A. Clustergram of the average phenoprints of the “small and starving” Keio deletion strains associated with an enrichment of over- (up) or under- (down) represented metabolites as defined by the z-scores in the dataset from Fuhrer et al., 2017. Over- and under-represented metabolites are defined as metabolites associated with a z-score above 2, or below -2, respectively. The enrichment analysis was performed as for Fig 4E (see Materials and methods). Only –H+ ions were included in this analysis for representation purposes. B. Clustergram of the average phenoprints of the “thin” Keio deletion strains associated with an enrichment of over- (up) or under- (down) represented metabolites as defined by the z-scores in the dataset from Fuhrer et al., 2017. Over- and under-represented metabolites are defined in A. Only –H+ ions were included in this analysis for representation purposes.

The way the datasets and the results (i.e. phenotypic clusters) are made available and made easily searchable for the community, could be improved.

We have submitted our data to the publicly available and easily searchable site, EcoliWiki. In addition, we improved the format of the data in the Excel files provided as supplementary files. The data in these Excel files are formatted as Tables, which means that wherever the reader is in the list of strains, the titles of the columns are still visible. In addition, filters are readily available by clicking on the arrows next to the

title of each column. The filters, for example, allow the reader to sort all the lines in the Table based on the values in one column. Finally, we also visually enhanced the readability of the scores in the tables by applying a color gradient filling the cells of the Table, from blue ($s = -6$) to red ($s = +6$). Any value in between -3 and 3 (i.e., below our score threshold) results in a white filling of the cell.

Reviewer #3:

The manuscript from Campos et al. describes an image-based screen for the *E. coli* single-gene knockout library, in which they quantified cellular morphologies, growth, and cell cycle-related properties, and then connected the quantitative phenotypical observations with functional analysis to associate the phenotypes with gene functionality. They additionally studied the correlation across genetic knockouts for cell dimensions, growth, and nucleoid dynamics, which revealed some systematic relationship across those metrics.

Overall, the manuscript is clearly written and the experimental data provide good resources on single-cell phenotypes across genetic knockouts. However, this manuscript presents system-scale data without extracting general rules or validating their discoveries with specific examples, especially for Figures 2 to 5. I would like to see some more concrete biological conclusions from all the statistics presented.

Although the first part of our manuscript is inevitably descriptive and provides an overview of our experimental setup and overall findings, we do not fully agree with the reviewer's comments. Our image-based, genome-wide screen uncovers a large number of genes associated with *E. coli* morphology, growth and cell cycle progression. Subsequent high-dimensional classification using tSNE allowed us to identify clusters sharing similar phenotypic alterations which, in turn, led to the identification of gene functions and pathways associated with cell size, shape and cell cycle dynamics (e.g. the ECA biosynthesis gene deletions affecting cell width control). During this process, we highlighted interesting mutants and aimed to provide a more mechanistic understanding of their corresponding phenotypes (e.g., strains in cell morphology island 22 associated with filamentation phenotypes such as *uup* or *rdgB*, and the identification of Δ *mraZ* as a potential gain-of-function cell size homeostasis mutant). We also provided interesting examples of deletions with unexpected and more difficult to rationalize phenotypes (e.g., the deletion of *pstACS* and *atpB*, *atpFHA*, and *atpC* genes leading to a reduction in cell width). At the same time, we also demonstrated the existence of many novel, previously uncharacterized genes involved in cell cycle progression (e.g. strains comprising cell cycle islands 4, 8 and 12).

More to the reviewer's point, our large high-dimensional dataset enabled us to examine, for the first time, global effects and dependencies between morphological, cell cycle and growth phenotypes, which, in turn, provides new insights into the potential coordination of these cellular processes. Using an information-theoretic Bayesian network analysis, we uncovered connections (or lack thereof) between phenotypic features which lead to a number of compelling and general biological conclusions. 1) Growth rate, for example, displayed no connectivity to morphological or cell cycle features, contrary to common belief. 2) The absence of correlation between cell length and cell width suggest that these cellular dimensions are independently regulated, implying that, at least under these growth conditions, cells do not control their size by monitoring their volume, surface area or the ratio between the two, as previously proposed for *Caulobacter crescentus* (Harris et al, Mol Microbiol, 2014). 3) Cell area and nucleoid area displays a remarkably strong positive correlation, regardless of the number of nucleoids per cell. This

scaling property draws a striking parallel with the 100-year-old observation that nucleus size scales with cell size in eukaryotes (Conklin, *J Exp Zool* 1912), an empirical relationship that has been reported for many eukaryotic cell types since (Vukovic et al, *Int Rev Cell Mol Biol* 2016). This suggests a universal size relationship between DNA-containing organelles and the cell across taxonomic kingdoms, even for organisms that lack a nuclear envelope. 4) The nucleoid size is negatively correlated with the relative timing of nucleoid segregation across 4,000 genetic perturbations, uncovering a new relationship between the dynamics of the bulk of chromosomal DNA and cell size. 5) Our network analysis (Fig 8) enabled us to go beyond pairwise correlations by integrating the complex set of interdependencies between cell morphogenesis, growth and cell cycle events. The structure of the constructed network indicates that the cell cycle is connected to cell morphogenesis through the size of the nucleoid (i.e., global structure of the chromosome).

Together, these observations suggest, for the first time, a potential route for establishing coordination between critical cellular processes and highlight the central role of the nucleoid herein.

In addition, I believe there are several issues on both experimental designs and data interpretation that need to be addressed.

1. A general comment: all the experiments in this manuscript were performed in M9 media with casamino acids and glucose at 30 °C. In such relative poor media with low temperature, cells grow slower and may be under quite different physiological states compared to other experiments where cells grown in LB, and/or experiments performed at 37 °C. For instance, multi-fork replication may not occur in the current condition, which means that cells may regulate DNA replication quite differently. Therefore, unless the authors can provide evidences that their discoveries still hold true for fast-growing conditions, it would be good to clearly state the differences in experimental conditions for all the conclusions made in the manuscript.

We agree with the reviewer that our observations apply to the tested growth conditions. We mentioned this reservation in the previous version of the manuscript (lines 378-385), but, as correctly pointed out by the reviewer, failed to report on the cell physiology that the current growth conditions give rise to. To address this issue, we performed a series of replication run-out experiments, which allow the quantification of ongoing DNA replication cycles. These experiments demonstrate the presence of overlapping replication cycles under our growth conditions. These results are now described lines 52-54 and 571-574, and presented in Fig S1A.

The reason for the overlapping replication cycles is because the growth medium is relatively rich (glucose is a preferred carbon source, and the medium was supplemented with casamino acids and thiamine). Temperature affects the rate of biochemical processes and can alter growth rate without affecting cell size and DNA replication patterns (Schaechter et al, *J Gen Microbiol* 1958, Shehata et al, *J Bacteriol* 1975, Stokke et al, *PLoS One* 2012). Lowering the temperature (within a reasonable window) thus allows one to create a slow-motion rendering of what occurs at higher temperatures. Together, these factors explain the presence of overlapping replication cycles in spite of the relatively low growth rates. We have revised the text to explain the growth conditions lead to multifork replication, which was important to clarify. Thank you for bringing this omission to our attention.

2. Due to the strong dependence of cellular morphology to experimental conditions, it is very critical to compare all the strains under the same conditions. One concern is that since different strains have different maximum OD, then the OD at which imaging is performed (~0.2) does not necessarily guarantee

exponential growth, especially for those with low maximum OD. So, additional controls are needed for these strains to confirm that morphology and cell cycle features are still independent of OD at which imaging is performed.

We agree. The OD range that corresponds to exponential growth does indeed vary for different mutants, depending on their growth capacity within the given nutritional environment. This is the reason why we consider the “OD at growth saturation” (OD_{max}) as an important phenotypic feature in our analysis. High-dimensional tSNE classification, which incorporates this information, leads to the clustering and identification of mutants displaying such growth defects (islands 1-4 and 21 in Fig. 4B). Although 12 strains (<0.3% of our dataset) were sampled at an OD_{600nm} > 50% of their OD_{max} (i.e., not at true steady state), the power of our approach lies in our ability to identify and cluster these cases.

In addition, we added panel D to Fig S1 to highlight that the vast majority of the Keio strains were imaged at an optical density much lower than the OD at saturation. The text has been revised to provide this information.

We have also verified that the OD_{600nm} window (0.091-0.355) across which cells were sampled did not introduce any detectable biases in morphological attributes (see figure below).

Figure. Sampling OD did not affect cell morphology. Scatter plots illustrating that the OD window across which cells were imaged in our screen did not introduce any significant biases in quantification of cell morphological features. The red dots and bars show the mean and standard deviation per individual bin, respectively. Each grey dot represents an individual strain (n = 4467).

3. Also, relating to the above issue about experimental conditions. Presumably imaging 48 strains on a large agar pad takes tens of minutes, during which cells would grow on the pad. Such growth may alter cell dimensions, and introduce newly synthesized unlabeled DNA. Thus, the bias introduced by imaging sequence should be addressed before further analysis.

We agree, which is why we examined if the elapsed time on the pad may affect any morphological or cell cycle features (Fig S2B and S3). Any identified biases were corrected before further analysis. Please see (Shi et al, Nat Prot 2017) for a similar example. We opted to use live cells in combination with spatio-temporal correction procedures because methods that use fixatives typically alter cell and nucleoid morphology (Chao et al, Appl Microbiol Biotechnol 2011, Liu et al, Scanning 2012). This information was provided in the original manuscript. However, to more clearly illustrate that these considerations were taken into account, we have added an additional supplemental figure (Fig S4) that illustrates the use and effect of our correction procedures.

4. In Fig 7A, there are many strains that have WT-like lengths but larger length CVs, which also indicates a mis-regulation in cell size homeostasis. Have the authors looked at any strains in that region?

Some strains with a WT-like mean length score and a high CV_L are deleted for genes directly related to DNA transactions or the SOS response (*recA*, *recC*, *holC*, *xerD*, *ruvC*, *fis*, *pcnB*, etc.). Defects in DNA repair or delays in DNA transactions can lead to cell division defects (Mulder et al, J Bacteriol 1989), which we mention in the text. Other mutants with high CV_L and WT-like mean length include strains deleted for ECA, CA or other cell envelope-related genes (*cpsG*, *wcaI*, *rseA*, *ompC*, *gmm*, *rfaP*, *arnC*). For these mutants, the cell length variability is likely a result of a general cell envelope stress. Other mutants have a high CV_L because of a high variability in the positioning of the division site (CV_{DR}). As we mentioned in the text, this class of cell division mutants can be identified by their abnormal CV_{DR} values.

Some strains with high CV_L but normal $\langle L \rangle$ and CV_{DR} have deletions in γ -genes. These uncharacterized genes may indeed play a role in cell size homeostasis, and would be interesting candidates to pursue in future studies.

5. Lines 416-421: the authors made a strong statement that since no correlation in length/width was observed, cells are not controlling their sizes by monitoring volume or surface area. However, since many of the genetic knockouts can directly affect volume growth and/or surface area growth (e.g. if cell wall incorporation or PG precursor synthesis is affected, cells are likely to reduce surface area to compensate; a similar effect is reported by Harris and Theriot, Cell 2016), then it is almost expected that volume or surface area cannot be conserved across genetic backgrounds. Therefore, the authors don't have enough evidence to conclude that mean length and width are independently regulated.

We do not exclude the possibility that some gene deletions directly affect volume or surface area growth. The power of our approach, however, lies in the great number and variety of genetic perturbations which allowed us to average out specific effects associated with each individual deletion. It also prevents potential biases caused by deletions that affect co-varying phenotypes independently. Please see Sachs K et al., Science 2005 and Collinet et al, Nature 2010 for similar examples in the eukaryotic literature. For a review on the topic, please see Liberali et al, Nat Rev Genet 2015. This comprehensive approach directly contrasts with that employed in previous studies which typically rely on a limited number of perturbations (Harris and Theriot, Cell 2016 and Zheng et al, PNAS 2016). In addition, we provide a wide-array of different perturbation types (covering almost all cellular processes) instead of only perturbing PG biosynthesis or a single protein involved in cell division or cell growth. The absence of a correlation between mean cell length and width across this broad spectrum of perturbations is what allowed us to conclude that length and width are independently regulated. We

have expanded the text to further clarify the reasoning underlying our approach and to include references (lines 420-440). In addition, to better explain the importance of sample size, we have also added concrete examples of how small sample size can be misleading and how increasing the sample size increases the confidence in our calculated correlation values (lines 454-457, new Fig EV5). In addition, we have added 95% confidence intervals to all provided correlations to show quantitatively the high confidence associated with our estimations. Finally, we added that our conclusion stands at least for the growth conditions examined, as the result may be different in different growth media.

Some minor points:

1. In Fig 2, the authors make the reference by using a Gaussian distribution. Would it make more sense to use the distribution of wild-type controls?

We do use the distribution of WT controls (no Gaussian fitting). To avoid confusion, we have clarified this point in the new version of the manuscript.

2. In Fig 3, category H is highly enriched for morphological features with scores ≤ 3 . Do the authors have any comments on that?

The strong enrichment for category H is driven by genes involved in pantothenate biosynthesis (e.g., *panE*), siderophore transport (e.g., *fepC*), the biosynthesis of electron carriers such as menaquinone and ubiquinone (e.g., *menB*, *ubiX*), biotin biosynthesis (e.g., *bioA*, *bioF*), or by genes related to chorismate biosynthesis (e.g., *pabC*). This COG category H encompasses a number of pathways central to the metabolism of cells in general. The enrichment of category H among small mutants suggests that, in general, cell size is affected by impairment in the metabolism and transport of coenzymes in a manner similar to nutritional restriction. We have revised the manuscript to add this point (lines 153-156).

3. Fig 7B, p value is needed for the comparison.

The statistical analysis is now mentioned in the legend of figure 7. The data presented on Fig 7B were a compilation of two experiments providing a total of 1,664 WT cells and 2,198 *ftsA** cells. Performing a Kruskal-Wallis multi-comparison test on the four sets of constriction degree values (2 for WT and 2 for *ftsA**) resulted in a p-value of 0.023. Bonferroni corrected post-hoc pairwise tests did not allow the distinction between WT and *ftsA** samples as only one post-hoc test revealed a significant difference at a threshold of 0.05 (p-value = 0.02) between one of the two WT samples and one of the two *ftsA** samples. All the other pairwise tests did not reveal any statistical differences. Moreover, the two WT samples were more “different” between each other than with the *ftsA** samples, precluding any clear statistical inference on the difference between the *ftsA** mutant and its parental strain.

Kruskal-Wallis table

Source	SS	df	MS	Chi-sq	Prob>Chi-sq
Groups	1.08E+07	3	3.61E+06	9.5184	0.0231
Error	4.38E+09	3858	1.14E+06		
Total	4.39E+09	3861			

Pot-hoc test - Bonferroni correction

Group1	Group2	$\Delta\mu$ 95%IC lower bound	$\Delta\mu$	$\Delta\mu$ 95%IC upper bound	p-value
WT1	WT2	-250.7762	-99.9939	50.7885	0.4811
WT1	ftsA*1	-316.6098	-151.0898	14.4302	0.0962
WT1	ftsA*2	-306.7657	-161.1142	-15.4628	0.0211
WT2	ftsA*1	-186.06	-51.096	83.8681	1
WT2	ftsA*2	-170.8144	-61.1204	48.5737	0.8494
ftsA*1	ftsA*2	-139.2309	-10.0244	119.1821	1
$\Delta\mu = \text{mean Grp1} - \text{mean Grp2}$					

Thank you for sending us your revised manuscript. We have now heard back from the two referees who were asked to evaluate your study. As you will see below, reviewers #1 and #3 still raise some remaining concerns, which we would ask you to address in a revision.

I think that the points raised by reviewer #3 are quite clear. Regarding point #1 of reviewer #1, we think that it can be addressed by providing explanations/clarifications in the text. Regarding point #2 of reviewer #1, we agree that further analyses of some 'width mutants' would enhance the impact of the study. However, since in the revised version of the manuscript the statements related to the implications of these findings (independent control of width and length) have been removed from the abstract we think that the addition of such experiments is not mandatory for the acceptance of the study for publication.

REFEREE REPORTS

Reviewer #1:

I still do not think its a great paper and the authors have not done much work to address my previous concerns. For example, just because many papers have derived cell cycle progression from snapshots in the past it does not necessarily make it right. As far as I understand the cell cycle progression model that is used assumes that the cell cycles are identical for cells of the same genotype disregarding the cell-to-cell variability in generation times and division sizes that clearly exists. The authors should at least explain the assumptions in the model and the possible consequences of these being incorrect.

Similarly, I did not ask the authors to test the adder model with time lapse imaging for all strains but only for "some different width mutants", which is more reasonable than doing it for all stains.

Overall the paper still represents a a big effort and it is valuable as resource, which could motivate publication.

Reviewer #2:

I sincerely apologize that it took me some time to re-review this manuscript. The authors have done a great deal of work in revising their manuscript, and have made the best from the Reviewer's input. I recommend acceptance of the manuscript.

Reviewer #3:

In this revision, the authors have addressed several concerns I raised previously by conduction extra control experiments and/or analysis, which have strengthened the conclusions they make. However, there are additional things that need to be analyzed to ensure the good quality of the data.

1. In response to my comment #2 (additional controls are needed for the strains with low maximum OD to confirm that morphology and cell cycle features are still independent of OD at which imaging is performed), the authors plotted a series of scatter plots showing that across the whole library, sampling OD did not systematically affect cell morphology. While this is informative, given the recent discovery that cell dimension changes rather rapidly during growth even at relatively low OD (Colavin et al, Nat Comm 2018), it would be good to also plot the same scatter plots for only wild-type cells under the experimental condition the authors have used.

2. In Fig S2A, looks like the mean width and CV width plots drop to certain extents for the plates imaged later. Did the authors correct for that? Also, the positional effect on a multiwell plate needs to be analyzed in a similar way (e.g. whether wells on the corner like A1 always have a systematic bias in morphology or growth). Although some of the biases are inevitable in such a large screen, it

would be informative to address them, which informs the community and helps the reproducibility of such datasets.

2nd Revision - authors' response

28th May 2018

We once again thank the editor and reviewers for their feedback and thoughtful comments. We have addressed their concerns and suggestions; please see below for a point-by-point response (in blue).

Response to editorial comments

Thank you for sending us your revised manuscript. We have now heard back from the two referees who were asked to evaluate your study. As you will see below, reviewers #1 and #3 still raise some remaining concerns, which we would ask you to address in a revision.

I think that the points raised by reviewer #3 are quite clear. Regarding point #1 of reviewer #1, we think that it can be addressed by providing explanations/clarifications in the text. Regarding point #2 of reviewer #1, we agree that further analyses of some 'width mutants' would enhance the impact of the study. However, since in the revised version of the manuscript the statements related to the implications of these findings (independent control of width and length) have been removed from the abstract we think that the addition of such experiments is not mandatory for the acceptance of study for publication.

We have addressed reviewer #3's points by providing two additional figures (please see below). The first point of reviewer #1 was addressed with textual clarification, as suggested. We believe that the addition of the experiments suggested by reviewer #1 would be useful only if performed with many mutants, which would be a lot of work and, in our opinion beyond the scope of this study. We are therefore very thankful that the editor does not find these experiments mandatory.

On a more editorial level, we would ask you to address the following issues:

- Please provide a .doc file for the main manuscript text.

Done

- Please include a Data and Software availability section at the end of the Materials and Methods describing how/where the data and computer code produced in this study have been made available.

Done

- We would also ask you to make the imaging data from the screen available at the image data repository Image Data Resource (IDR) <https://idr.openmicroscopy.org> or at a 'general' repository e.g. Biostudies or Dryad. Please include the DOI of the dataset in the Data and Software availability section.

We are in the process of uploading and submitting our dataset to the BioStudies Database. The accession number for our dataset is S-BSST151.

- Please provide a "standfirst text" summarizing the study in one or two sentences (approximately 250 characters), three to four "bullet points" highlighting the main findings and a "thumbnail image" (211x157 pixels, jpeg format) to highlight the paper on our homepage.

Done

- Please rename Computer Code EV1A & B to Computer Code EV1 and EV2 and update the callouts in the manuscript file accordingly. The same for Computer code 1 and 2: please rename to Computer Code EV3 and EV4 and include a callout in the text.

We have renamed the computer codes accordingly. Computer code 1 and 2 stemmed from an older version of the manuscript and correspond to Computer Code EV1 and EV2.

- The text in the manuscript file describing the datasets and computer codes (p. 47 and 48) will not be included in the published paper. Please make sure that all information related to the respective files can be found within the actual datasets/computer code zip files.

We have removed this text from the manuscript file and ensured that all related information is available within the respective files.

Response to reviewer comments

Reviewer #1:

I still do not think its a great paper and the authors have not done much work to address my previous concerns. For example, just because many papers have derived cell cycle progression from snap-shots in the past it does not necessarily make it right. As far as I understand the cell cycle progression model that is used assumes that the cell cycles are identical for cells of the same genotype disregarding the cell-to-cell variability in generation times and division sizes that clearly exists. The authors should at least explain the assumptions in the model and the possible consequences of these being incorrect.

Although our approach does not consider single-cell level heterogeneity, we do not assume cell cycles to be identical for isogenic cells. Instead, we extract population-level-average cell cycle timings from static images by quantifying the fraction of cells not having undergone a given cell cycle event (Wold et al, Embo J, 1994). We would like to emphasize that this approach is not a model but rather a direct consequence of exponentially growing populations in steady-state (Kafri et al, Nature, 2013). The only assumption is that populations are growing in steady-state, which, we believe, is fair given that cells were sampled during the early stage of

population growth (i.e., low ODs). The cell cycle timings obtained in this way represent population level average timings that would also be obtained when averaging the behavior of many individual cells of that same population. In the previous rebuttal, we provided time-lapse data demonstrating the validity of this approach. Also, as stated in the previous rebuttal, other groups have not only used this approach, but have also validated it with time-lapse experiments, demonstrating that the method is correct.

We have expanded the revised version of the manuscript to further describe our methodology and explain the type of information it allows to extract in contrast to what it cannot extract.

Similarly, I did not ask the authors to test the adder model with time lapse imaging for all strains but only for "some different width mutants", which is more reasonable than doing it for all stains.

We agree with the reviewer that these would be interesting experiments to pursue. However, as we explained in the previous rebuttal, these experiments would only be conclusive if many different width mutants were included and other aspects besides their size expansion (e.g., DNA replication period) were measured. These experiments would represent an enormous amount of work, considering the time-consuming and labor-intensive nature of such experiments and the fact that a new microfluidic mold would have to be constructed for each mutant (microfluidic devices are sensitive to variation in cell width; cells that are too thin are washed away, whereas cells that are too thick don't get in or get jammed). Therefore, we believe that the suggested experiments fall beyond the scope of this study. We are grateful that the editor deems these experiments not necessary for publication.

Overall the paper still represents a a big effort and it is valuable as resource, which could motivate publication.

We thank the reviewer for his/her appreciation of the magnitude of our work and its value as a resource for the scientific community.

Reviewer #2:

I sincerely apologize that it took me some time to re-review this manuscript. The authors have done a great deal of work in revising their manuscript, and have made the best from the Reviewer's input. I recommend acceptance of the manuscript.

We thank the reviewer for his/her appreciation of our work.

Reviewer #3:

In this revision, the authors have addressed several concerns I raised previously by conduction extra control experiments and/or analysis, which have strengthened the conclusions they make. However, there are additional things that need to be analyzed to ensure the good quality of the data.

1. In response to my comment #2 (additional controls are needed for the strains with low maximum OD to confirm that morphology and cell cycle features are still independent of OD at which imaging is performed), the authors plotted a series of scatter plots showing that across the whole library, sampling OD did not systematically affect cell morphology. While this is informative, given the recent discovery that cell dimension changes rather rapidly during growth even at relatively low OD (Colavin et al, Nat Comm 2018), it would be good to also plot the same scatter plots for only wild-type cells under the experimental condition the authors have used.

Per the reviewer's request, we have verified that the OD_{600nm} window (0.11-0.27) across which wild-type cells were sampled did not introduce any systematic biases in morphological or cell cycle attributes (see figure below).

Figure. Scatter plots illustrating that the OD window across which WT replicates were imaged did not introduce any significant systematic biases in the quantification of cell morphological or cell cycle features ($n = 240$ WT replicates).

2. In Fig S2A, looks like the mean width and CV width plots drop to certain extents for the plates imaged later. Did the authors correct for that? Also, the positional effect on a multiwell plate needs to be analyzed in a similar way (e.g. whether wells on the corner like A1 always have a systematic bias in morphology or growth). Although some of the biases are inevitable in such a large screen, it would be informative to address them, which informs the community and helps the reproducibility of such datasets.

As detailed in the data analysis section of the materials and methods, we did correct for plate-by-plate variability. For each plate, this was accomplished by setting the median values of each feature to the median feature value of the parental strain.

In addition, we have verified that positional effects from the multiwell plates did not introduce any detectable and/or systematic biases by averaging feature behavior of strains located in the same well across the 45 plates that comprise the Keio

collection (see figure below). Although apparent biases were evident in the raw data (see Figure below, panel A), these biases likely reflect temporal biases introduced during imaging (as plates were consistently imaged in the same order), and disappeared completely after correction and normalization in the scores (see figure below, panel B).

Figure. Graphical representation of 96-well plates for all morphological and cell cycle features displaying average feature behavior of strains located in a given well (averaged across all 45 plates that comprise the Keio collection) for the (A) raw data and (B) normalized scores. Each well is color-coded based on the average feature behavior of that well with the color scale centered on the mean value of the feature and extending 2 standard deviations away from the mean in each direction. Biases apparent in the raw data disappeared after correction and normalization.

Corresponding Author Name: Christine Jacobs-Wagner

Manuscript Number: MSB-17-7573